# Mitigating Local Cohesion and Global Sparseness in Graph Contrastive Learning with Fuzzy Boundaries

Yuena Lin [* 1]   Haichun Cai [* 2]   Jun-Yi Hang [3]   Haobo Wang [4]   Zhen Yang [1]   Gengyu Lyu [† 1]

## Abstract

Graph contrastive learning (GCL) aims at narrowing positives while dispersing negatives, often causing a minority of samples with great similarities to gather as a small group. It results in two latent shortcomings in GCL: 1) *local cohesion* that a class cluster contains numerous independent small groups, and 2) *global sparseness* that these small groups (or isolated samples) dispersedly distribute among all clusters. These shortcomings make the learned distribution *only focus on local similarities among partial samples, which hinders the ability to capture the ideal global structural properties among real clusters, especially high intra-cluster compactness and inter-cluster separateness*. Considering this, we design a novel fuzzy boundary by extending the original cluster boundary with fuzzy set theory, which involves fuzzy boundary construction and fuzzy boundary contraction to address these shortcomings. The fuzzy boundary construction dilates the original boundaries to bridge the local groups, and the fuzzy boundary contraction forces the dispersed samples or groups within the fuzzy boundary to gather tightly, jointly mitigating local cohesion and global sparseness while forming the ideal global structural distribution (Zhang et al., 2024). Extensive experiments demonstrate that a graph auto-encoder with the fuzzy boundary significantly outperforms current state-of-the-art GCL models in both downstream tasks and quantitative analysis.

---

[*]Equal contribution  [1]College of Computer Science, Beijing University of Technology, Beijing, China [2]School of Computer and Data Science, Fuzhou University, Fuzhou, China [3]School of Computer Science and Engineering, Southeast University, Nanjing, China [4]School of Software Technology, Zhejiang University, Hangzhou, China. Correspondence to: Gengyu Lyu <lyugengyu@bjut.edu.cn>.

*Proceedings of the 42$^{nd}$ International Conference on Machine Learning*, Vancouver, Canada. PMLR 267, 2025. Copyright 2025 by the author(s).

## 1. Introduction

Recent years have seen the rapid development of self-supervised graph representation learning (SSGRL) (You et al., 2021), which expects to extract critical features by mining intrinsic characteristics from raw graph data, such as commonalities and discriminalities (Wang & Liu, 2022; Jiang et al., 2021). Graph contrastive learning (GCL) as one of the most well-known SSGRL paradigms, learns discriminalities by dispersing negatives and mines commonalities by closing the positives (Xu et al., 2024a; Zou & Liu, 2023). Unfortunately, the learned distribution in the embedding space *only reflects local similarities within each cluster, and fails to showcase global structural distribution with high intra-cluster compactness and inter-cluster separateness*. As a matter of fact, the problem can be traced to two ingrained shortcomings in GCL, namely **local cohesion** and **global sparseness**.

To shed light on the shortcomings intuitively, we present the t-SNE visualization (Van der Maaten & Hinton, 2008) from three cornerstone models in Figure 1, where DGI (Velickovic et al., 2019) views nodes in the raw graph as positives and those in the corrupt graph are negatives; GRACE (Zhu et al., 2020) treats corresponding nodes between two congruent views as positives while other nodes are negatives; BGRL (Thakoor et al., 2022) selects positives in the same way as GRACE but scatters the negatives in a latent fashion. The t-SNE of these models reveals the two common shortcomings in GCL: 1) **Local cohesion results in numerous small groups**. The sample selection strategies in GCL tend to gather a minority of samples with great similarities, which leads to numerous independent small groups in a real cluster. 2) **Global sparseness across cluster distributions**. These small groups (or isolated samples) are discretely distributed in each cluster, which only occupy part locations while leaving considerable blank space in each cluster. The crux of the shortcomings may be attributed to the fact that **existing GCL models have no access to label supervision for building reliable boundaries**. *Unreliable boundaries, such as excessive boundaries that incur samples from other clusters, or limited boundaries that are unable to involve sufficient samples from the same cluster, are harmful to the structural distribution* (Chen et al., 2023).

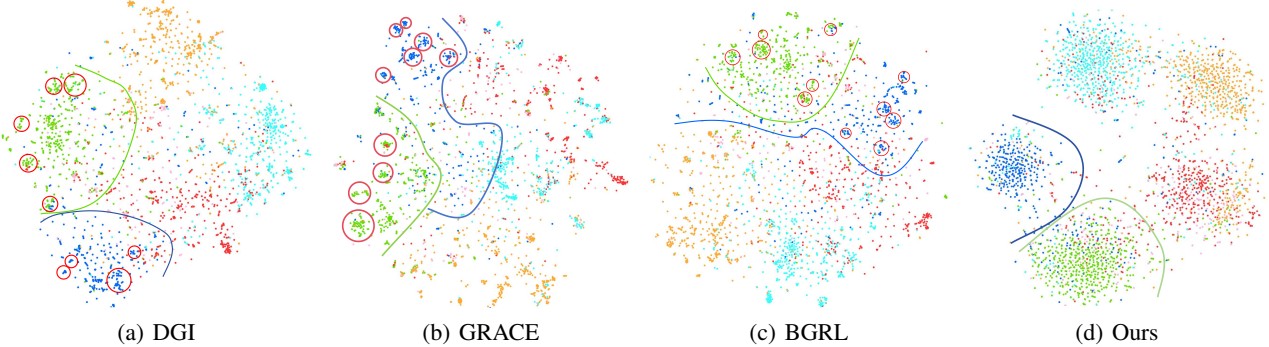

| (a) DGI | (b) GRACE | (c) BGRL | (d) Ours |

*Figure 1.* t-SNE for our model and three representative GCL models on the Citeseer dataset, i.e., DGI, GRACE, and BGRL. The boundaries for green and blue clusters are outlined. The two clusters exhibit global sparseness and local cohesion that forms numerous small groups (shown in red circles). Besides, near the boundary, the samples from the two clusters are interwoven together.

Boundaries are closely related to the aforementioned shortcomings but are seldom taken into consideration in GCL. Considering this, we design a novel learnable boundary called fuzzy boundary by fusing fuzzy set theory to extend the original clusters for addressing the local cohesion and global sparseness. The fuzzy boundary is concerned with fuzzy representations and crisp representations, where the fuzzy representations are the learned representations applied with fuzzy set theory, or crisp representations otherwise. The proposed fuzzy boundary includes the fuzzy boundary construction phase and the fuzzy boundary contraction phase. In the fuzzy boundary construction phase, we transform the crisp representation space to the fuzzy representation space with fuzzy set theory, where we expand the original cluster boundaries as the new-born fuzzy boundaries to further bridge the local groups through learning the fuzzy positives of crisp representations. When turning the fuzzy boundary contraction phase, we narrow the fuzzy boundaries and the corresponding cluster prototypes while pushing the fuzzy boundaries away from other cluster prototypes, effectively tightening the dispersed samples or groups within the fuzzy boundaries. Finally, the local groups or isolated samples within each cluster become interconnected and the ideal global structural distribution with high intra-cluster compactness and inter-cluster separateness gradually emerges, forming more distinct boundaries. Our contributions are summarized as follows.

- We first attend to two ingrained shortcomings causing the inferior global structural distribution in GCL: local cohesion and global sparseness, and disclose that these shortcomings suffer from unreliable boundaries.

- We propose a novel fuzzy boundary to bridge the discrete local groups and isolated samples, and tighten each cluster, which effectively alleviates the local cohesion and global sparseness, and realizes ideal global structural distributions.

- We implement the fuzzy boundary on a pre-trained graph auto-encoder (GAE), which makes the model performance enhanced and comparable with current state-of-the-art SSGRL models in the downstream tasks and quantitative analysis.

## 2. Related Work

### 2.1. Fuzzy Set Theory

Fuzzy set theory generalizes classical set theory that determines whether an object belongs to a set, and incorporates the idea that elements have membership degrees in a set (Zadeh, 1965). Here, we introduce the critical definitions of the fuzzy set theory in the following:

**Definition 2.1.** A universe $\mathbb{U}$ is defined as a set of all possible terms occurring in a certain domain.

**Definition 2.2.** A function $f_{\mathcal{A}}: \mathbb{U} \longrightarrow \mathbb{L} \subseteq \mathbb{R}$ is called a membership function.

**Definition 2.3.** A pair composed of the universe and the membership function $\mathcal{A} = (\mathbb{U}, \ f_{\mathcal{A}})$ is called a fuzzy set.

A fuzzy set $\mathcal{A}$ in the universe $\mathbb{U}$ is characterized by a membership function $f_{\mathcal{A}}(x)$ which associates each point in $\mathbb{U}$ with a real value in $[0, 1]$. The value of $f_{\mathcal{A}}(x)$ represents the membership degree of $x$ in $\mathcal{A}$. The fuzzy set theory provides a powerful framework for reasoning about sets with uncertainty, but the specification of membership functions depends on the domain that the objective task is located in (Zhelezniak et al., 2018). Following the definitions, when $\mathbb{L} = [0, 1]$, $\mathcal{A}$ is a fuzzy set. When the membership function takes on only two values 0 and 1, namely $\mathbb{L} = \{0, 1\}$ according to whether $x$ belongs to $\mathcal{A}$, in this case, $\mathcal{A}$ reduces to a classical set, also called a crisp set.

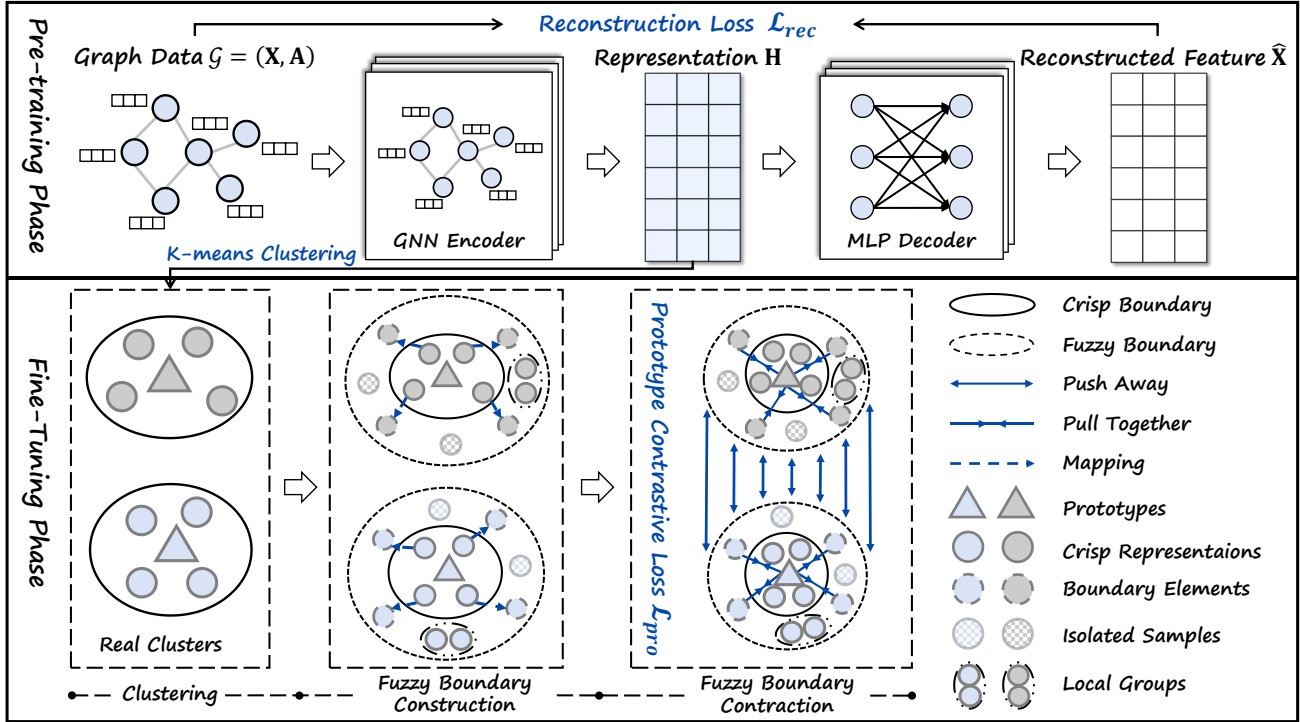

Figure 2. Overview of the proposed model. Firstly, the GNN encoder is pre-trained with the reconstruction loss $\mathcal{L}_{rec}$. Secondly, the output representation **H** is used to obtain real clusters through the K-means clustering algorithm. Then the original boundaries (in solid lines) are extended as fuzzy boundaries (in dashed lines) to incorporate isolated samples and local groups. Finally, the fuzzy boundaries are contracted to tighten samples within the fuzzy boundaries while separating different clusters by prototypical contrastive loss $\mathcal{L}_{pro}$.

## 2.2. Self-Supervised Graph Representation Learning

Exempt from the supervision of downstream tasks, self-supervised graph representation learning (SSGRL) focuses on extracting generalized and invariant features from the raw graph data. Early works were mainly based on random walk, which only took structure information into consideration. Subsequent works integrated feature information and turned to deep learning methods because of their tremendous fitting power to approximate complicated functions (Lv et al., 2024). Deep SSGRL models are roughly categorized into GAEs (Hou et al., 2022), graph generative adversarial networks (Wang et al., 2019), and graph contrastive models. Among them, graph contrastive models stand out for the simple-yet-effective concept that narrows positive pairs while dispersing negative pairs to maximize the lower bound of mutual information (Hu et al., 2024; Zhang et al., 2023a). Despite showing competitive performance, existing GCL models still exhibit two shortcomings, i.e., local cohesion and global sparseness, which cause inferior global structural distribution but are never addressed from the perspective of boundaries. Actually, good boundaries alleviate these shortcomings and form a reliable global structural distribution, which becomes the motivation of the proposed model.

## 3. Method

The proposed model includes a pre-training phase and a fine-tuning phase, where the pre-training phase includes a GAE framework to train a GNN encoder, and the fine-tuning phase contains the fuzzy boundary construction process and the fuzzy boundary contraction process for fine-tuning the GNN encoder. The output representation **H** of the pre-trained GNN is implemented on the K-means clustering algorithm to attain reliable real clusters, which is used in the fine-tuning phase. The whole architecture is shown in Figure 2. To provide details of the model for convenience, we introduce the following notations.

**Notations.** In this paper, a graph is denoted as $\mathcal{G} = (\mathcal{V}, \mathcal{E})$, where $\mathcal{V} = \{v_1, v_2, \cdots, v_N\}$ is the node set with $N$ nodes and $\mathcal{E} \subset \mathcal{V} \times \mathcal{V}$ is the edge set. The node attribute matrix and adjacency matrix are denoted as $\mathbf{X} \in \mathbb{R}^{N \times D}$ and $\mathbf{A} \in \{0, 1\}^{N \times N}$ respectively. $\mathbf{x}_i$ represents the attribute of node $v_i$, and $\mathbf{A}_{ij} = 1$ describes a relation between $v_i$ and $v_j$, or $\mathbf{A}_{ij} = 0$ otherwise. Our purpose is to train an $L$-layer graph encoder $f : \mathbb{R}^{N \times D} \times \{0, 1\}^{N \times N} \to \mathbb{R}^{N \times d_L}$ to produce low-dimensional node representations via raw attribute and structure contents, where $d_L$ is the dimension of the $L$-th layer, $D$ is the dimension of the raw attribute and $d_L \ll D$.

## 3.1. Pre-Training Process

To obtain relatively reliable clusters for constructing the fuzzy boundaries, we adopt a GAE (Thomas & Welling, 2016) for pre-training. During the pre-training process, we reconstruct the attribute matrix with $\mathcal{O}(ND)$ complexity instead of the adjacency matrix with $\mathcal{O}(N^2)$ complexity, which is defined as

$$\mathbf{H} = GNN(\mathbf{X}, \mathbf{A}), \hat{\mathbf{X}} = MLP(\mathbf{H}), \mathcal{L}_{rec} = \left\| \mathbf{X} - \hat{\mathbf{X}} \right\|_F, \tag{1}$$

where $\hat{\mathbf{X}}$ is the reconstructed attribute matrix, and $\|\cdot\|_F$ is $F$-norm. Since $D \ll N$ holds in general, restoring the embeddings to the raw attributes saves much more overhead than reconstructing the adjacency matrix. To avoid the over-smoothing problems caused by numerous GNN layers (Rong et al., 2019), we employ an asymmetric GAE backbone where GNNs and multi-layer perceptrons (MLPs) are implemented on encoding and decoding processes respectively as (Xiao et al., 2022; 2023). Specifically, we adopt a graph convolutional network (GCN) (Kipf & Welling, 2017; Xu et al., 2023) as the GNN encoder, and the encoding and decoding processes are given by

$$\mathbf{H}^{(l)} = \sigma \left( \widetilde{\mathbf{D}}^{-\frac{1}{2}} \widetilde{\mathbf{A}} \widetilde{\mathbf{D}}^{-\frac{1}{2}} \mathbf{H}^{(l-1)} \mathbf{W}^{(l)} + \mathbf{b}^{(l)} \right),$$
$$\hat{\mathbf{H}}^{(t)} = \sigma \left( \hat{\mathbf{H}}^{(t-1)} \mathbf{W}^{(t)} + \mathbf{b}^{(t)} \right), \tag{2}$$

where $\mathbf{H}^{(l)}$ and $\hat{\mathbf{H}}^{(t)}$ are the output representation matrices from the $l$-th layer encoder and the $t$-th layer decoder respectively, $\mathbf{W}^{(l)}$, $\mathbf{W}^{(t)}$, $\mathbf{b}^{(l)}$, and $\mathbf{b}^{(t)}$ are trainable weight matrices or bias vectors, $\widetilde{\mathbf{D}}$ is a diagonal matrix with diagonal element $\widehat{\mathbf{D}}_{ii} = \sum_j \widehat{\mathbf{A}}_{ij}$, $\widetilde{\mathbf{A}} = \mathbf{A} + \mathbf{I}_N$. $\sigma(\cdot)$ is the non-linear activation function, and it is not implemented on the last layer of the MLP decoder.

The pre-training process unleashes the capability of the graph model to mine commonalities and discriminalities, and encourages to generate relatively reliable clusters for constructing the fuzzy boundary. For convenience, we centralize all the node representations by subtracting their corresponding prototypes $\mathbf{C}$ as $\mathbf{Z}$, and utilize $\mathbf{Z}$ to construct the fuzzy boundaries, where the corresponding prototypes $\mathbf{C}$ are obtained by applying the K-means clustering algorithm on the output representation $\mathbf{H}$ of the GNN encoder. To save the costs on the clustering process, we execute the K-means clustering algorithm every $T$ epochs.

## 3.2. Fuzzy Boundary

### 3.2.1. Fuzzy Boundary Construction

The proposed model transfers the fundamental concept that a crisp set can be viewed as a special type of fuzzy set into representation learning, where a crisp representation is regarded as a particular type of fuzzy representation, which

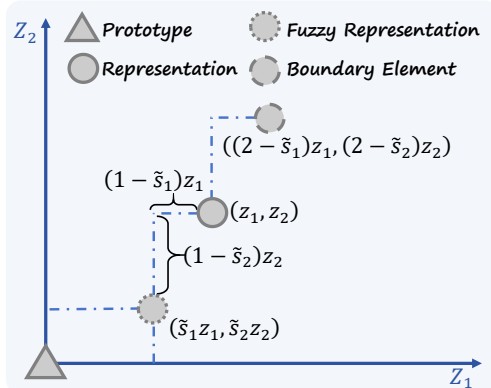

*Figure 3.* The construction process of fuzzy boundary in 2-dimension space. For one specific representation ◯, it is firstly utilized to calculate the fuzzy representation ⬤, which is then mapped to be the element of the fuzzy boundary ◯.

is utilized to find the appropriate elements of the fuzzy boundaries. These elements are expected to have two essential characteristics consistent with fuzzy representations. i) **Semantics-positive**. Since the constructed fuzzy boundary is assumed as a critical part of one real cluster, the elements scatted in the fuzzy boundary ought to share the commonalities with the samples within the real cluster, namely the elements of the fuzzy boundaries are the positives of the cluster samples. ii) **Learnable**. The fuzzy boundaries rely on the learnable membership functions, making the fuzzy boundaries adaptive to regulate the spectrum of a cluster in the training process. Among various membership functions, the Gaussian membership function is the most widely used (Zhang et al., 2023b; Li et al., 2024).

Generally, one Gaussian membership function is in accordance with one specific fuzzy semantics, and in this paper, we set $K$ membership functions to enrich the fuzzy semantics of the representations. For each value of one feature vector, its $k$-th membership degree is computed by

$$\mathbf{s}_{i,j}^k = exp \left\{ -\frac{\left( \mathbf{z}_{i,j} - \mathbf{m}_j^k \right)^2}{2 \left( \delta_j^k \right)^2} \right\}, j = 1, 2, \cdots, d, \tag{3}$$

where $\mathbf{z}_{i,j}$ is the $j$-th value of the centralized feature vector $\mathbf{z}_i$ and $\mathbf{s}_{i,j}^k$ is corresponding membership degree, $d$ is the dimension of $\mathbf{z}_i$, $\mathbf{m}_j^k$ and $\delta_j^k$ are the learnable mean and standard deviation of the $k$-th Gaussian membership function respectively. These membership degrees are then unified by a $Union$ operator to output the final one as

$$\tilde{\mathbf{s}}_{i,j} = Union \left\{ \mathbf{s}_{i,j}^k \right\}, k = 1, 2, \cdots, K. \tag{4}$$

The choice of the $Union$ operator is explained in Section 3.2.2, and the final membership degrees from the $Union$

operator define the fuzzy representations as

$$\widetilde{\mathbf{Z}} = \widetilde{\mathbf{S}} \odot \mathbf{Z}, \tag{5}$$

where $\widetilde{\mathbf{S}}$ is the membership degree matrix, $\mathbf{Z}$ is the centralized representation matrix, and $\widetilde{\mathbf{Z}}$ is the fuzzy representation matrix that rescales the crisp representations matrix $\mathbf{Z}$ via the adaptive membership degree matrix $\widetilde{\mathbf{S}}$. Crisp representations and fuzzy representations are assumed to be positives, thus the coordinate difference between them is deemed acceptable semantics difference between positives. Considering this, we employ the acceptable discrepancy to expand the cluster boundary, and the process is called fuzzy boundary construction, which is shown in Figure 3. The calculated coordinates of fuzzy representations, obtained through Eq. (5), are utilized to compute the corresponding boundary elements by

$$\overline{\mathbf{Z}} = \widetilde{\mathbf{Z}} + 2\left(\mathbf{1} - \widetilde{\mathbf{S}}\right) \odot \mathbf{Z} = \left(\mathbf{2} - \widetilde{\mathbf{S}}\right) \odot \mathbf{Z}, \tag{6}$$

where $\mathbf{1}$ and $\mathbf{2}$ are two real value matrices whose each element is 1 and 2 respectively, and $\overline{\mathbf{Z}}$ contains the elements of the fuzzy boundary. It is worth noting that the fuzzy boundary won't deteriorate into the crisp boundary, and it effectively connects the independent groups and isolated samples to mitigate the local cohesion.

**Theorem 3.1.** *Assuming that $N$ samples have different features, $K$ Gaussian membership functions have different means, and $K < N$, fuzzy boundaries never deteriorate into crisp boundaries.*

*Proof:* For Gaussian membership functions, the membership degree is 1 only when the feature element equals the mean. Suppose $\widetilde{\mathbf{S}} = \mathbf{1}$, which implies that the feature element must equal the mean of the Gaussian function. However, since the features of the $N$ samples are all different and the means of the $K$ Gaussian functions are also different, the means of the $K$ Gaussian functions can match at most $K$ feature elements. Therefore, it is impossible to obtain $\widetilde{\mathbf{S}} = \mathbf{1}$ and the fuzzy boundaries won't deteriorate.

### 3.2.2. Union Operator Determination

The fuzzy boundary elements are considered the critical part of the corresponding cluster, and the variances of the extended fuzzy clusters could reflect the compactness of the real clusters. Thus, the $Union$ operator is crucial to adjust the cluster variances for realizing the intra-cluster compactness and inter-cluster separateness.

**Theorem 3.2.** *When $Union$ operator adopts max-pooling, the overall variances within each cluster are minimized.*

The detailed proof is provided in **Appendix A.1**. Additionally, the $Union$ operator has another advantage, namely the distances between crisp representations and fuzzy representations are minimal in the representation space. According

to the fuzzy representation defined in Eq. (5), when we use the ordinary distance function $Dis\left(\cdot\right)$ such as Euclidean distance to measure the distances between one crisp representation and the corresponding fuzzy representation, we have the following conclusion as

$$\begin{aligned} Dis\left(\{\tilde{\mathbf{s}}_{min}\}_i \odot \mathbf{z}_i, \mathbf{z}_i\right) &\geq Dis\left(\{\tilde{\mathbf{s}}_{other}\}_i \odot \mathbf{z}_i, \mathbf{z}_i\right) \\ &\geq Dis\left(\{\tilde{\mathbf{s}}_{max}\}_i \odot \mathbf{z}_i, \mathbf{z}_i\right), \end{aligned} \tag{7}$$

where $\{\tilde{\mathbf{s}}_{min}\}_i$ is the membership degree vector of node $v_i$, which is filled with minimal membership degrees, and its element is $\{\tilde{\mathbf{s}}_{min}\}_{i,j} = \min_k\{\mathbf{s}_{i,j}^k\}$. Similarly, we have $\{\tilde{\mathbf{s}}_{max}\}_{i,j} = \max_k\{\mathbf{s}_{i,j}^k\}$. $\{\tilde{\mathbf{s}}_{other}\}_i$ denotes other simple $Union$ operators, such as *mean-pooling*. This inequation means that the *max-pooling* is in line with searching for the representations with maximal similarities in the fuzzy representation space, and also implies that samples inside the constructed fuzzy boundary are likely to be part of this class cluster with great (membership) degrees.

### 3.2.3. Fuzzy Boundary Contraction

During the fuzzy boundary contraction phase, it is hard to achieve the inter-cluster separateness in a centralized representation space since the centralized clusters are concentrated around the origin points, so we need to restore the coordinates of fuzzy representations as $\widetilde{\mathbf{H}}$ by adding their corresponding prototypes $\mathbf{C}$. Considering the restored fuzzy boundaries correspond to new-born fuzzy prototypes $\mathbf{F}$, and we need to balance $\mathbf{C}$ and $\mathbf{F}$ to obtain more ideal prototypes. A naive method like the convex combination to fuse the fuzzy prototype and the crisp prototype is always suboptimal in practice, and we apply an MLP $g\left(\cdot\right)$ to better balance the relationship of both as

$$\mathbf{P} = g\left(\mathbf{F} \parallel \mathbf{C}\right), \tag{8}$$

where $\mathbf{P} \in \mathbb{R}^{C_p \times d}$ is the fused prototype matrix, $C_p$ is the number of clusters, $\mathbf{C} \in \mathbb{R}^{C_p \times d}$ is the crisp prototype matrix obtained by applying clustering algorithms on representation $\mathbf{H}$, $\mathbf{F} \in \mathbb{R}^{C_p \times d}$ is the fuzzy prototype matrix by averaging the fuzzy boundary elements for each cluster, and $\parallel$ is the concatenation operator. To densify the cluster distribution and differentiate different clusters, we apply one prototype loss between the prototypes and fuzzy boundaries, which is defined as

$$\mathcal{L}_{pro} = -\frac{1}{N} \sum_{i=1}^{N} \log \frac{e^{-\tau \left\|\widetilde{\mathbf{h}}_i - \mathbf{p}_i\right\|_2^2}}{\sum_{j=1}^{C_p} e^{-\tau \left\|\widetilde{\mathbf{h}}_i - \mathbf{p}_j\right\|_2^2}}, \tag{9}$$

where $\mathbf{p}_i$ is the fused prototype that node $v_i$ belongs to and is computed by Eq. (8). $\widetilde{\mathbf{h}}_i$ is the restored fuzzy boundary elements, $\tau$ is the temperature coefficient, and higher $\tau$ weighs the intra-cluster compactness more. Through contracting each cluster, the samples or local groups in the

*Table 1.* Results in node classification. The best results are highlighted in bold, and the second-best results are highlighted with underline. "OOM" indicates that the graph models raise the out-of-memory failure.

| Dataset | Cora | Citeseer | Pubmed | DBLP | Ogbn-arxiv | Amazon Photo | Amazon Computers |
|---|---|---|---|---|---|---|---|
| DeepWalk | 74.6 ± 0.2 | 50.8 ± 0.1 | 80.1 ± 0.2 | 76.0 ± 0.7 | 63.6 ± 0.4 | 89.4 ± 0.1 | 85.7 ± 0.1 |
| DGI | 82.6 ± 0.4 | 68.8 ± 0.7 | 86.0 ± 0.1 | 83.2 ± 0.1 | 67.9 ± 0.4 | 91.6 ± 0.2 | 84.0 ± 0.5 |
| GMI | 83.8 ± 0.8 | 72.8 ± 0.8 | 85.5 ± 0.1 | 82.3 ± 0.2 | 67.1 ± 0.2 | 90.7 ± 0.2 | 82.2 ± 0.3 |
| GRACE | 83.3 ± 0.4 | 72.1 ± 0.5 | 86.7 ± 0.1 | 84.2 ± 0.1 | 67.4 ± 0.4 | 92.8 ± 0.5 | 89.5 ± 0.4 |
| GCA | 82.8 ± 0.3 | 71.5 ± 0.3 | 86.0 ± 0.2 | 83.1 ± 0.2 | 68.2 ± 0.2 | 92.2 ± 0.2 | 87.5 ± 0.5 |
| COAST | 84.3 ± 0.2 | 72.9 ± 0.3 | 86.0 ± 0.2 | 84.5 ± 0.1 | 62.1 ± 0.1 | 92.6 ± 0.5 | 88.3 ± 0.0 |
| AFGRL | 83.3 ± 0.9 | 71.5 ± 0.8 | 85.2 ± 0.2 | 83.1 ± 0.1 | 56.1 ± 0.1 | 93.2 ± 0.3 | 89.9 ± 0.3 |
| SUGRL | 85.2 ± 0.6 | 73.5 ± 0.6 | 86.7 ± 0.2 | 83.0 ± 0.2 | 68.7 ± 0.0 | 93.2 ± 0.4 | 88.9 ± 0.2 |
| iGCL | 84.0 ± 0.5 | 72.9 ± 0.9 | 86.2 ± 0.3 | 83.7 ± 0.2 | 65.6 ± 0.0 | 93.1 ± 0.3 | 90.1 ± 0.4 |
| NCLA | 85.3 ± 0.4 | 73.2 ± 0.5 | 85.5 ± 0.4 | 84.0 ± 0.2 | 58.4 ± 0.0 | 93.5 ± 0.2 | 89.1 ± 0.4 |
| GREET | 85.7 ± 0.5 | 73.3 ± 0.6 | 86.9 ± 0.3 | 83.8 ± 0.1 | OOM | 92.9 ± 0.3 | 87.9 ± 0.4 |
| HomoGCL | 85.4 ± 0.9 | 72.3 ± 0.4 | 86.3 ± 0.2 | 84.4 ± 0.2 | 62.5 ± 0.5 | 93.3 ± 0.2 | 89.2 ± 0.5 |
| S2GAE | 85.5 ± 0.3 | 73.4 ± 0.5 | 86.5 ± 0.1 | 84.1 ± 0.0 | 60.9 ± 0.0 | 93.5 ± 0.2 | 90.0 ± 0.1 |
| SGRL | 85.4 ± 0.2 | 73.2 ± 0.2 | 86.2 ± 0.0 | 84.1 ± 0.0 | 65.7 ± 0.0 | **93.8 ± 0.0** | 90.0 ± 0.0 |
| PiGCL | 84.6 ± 0.8 | 73.5 ± 0.6 | 86.3 ± 0.3 | 84.3 ± 0.3 | OOM | 93.1 ± 0.3 | 89.3 ± 0.3 |
| Bandana | 85.5 ± 0.8 | 73.0 ± 0.8 | 86.4 ± 0.3 | 83.9 ± 0.0 | 66.8 ± 0.0 | 93.4 ± 0.1 | 89.6 ± 0.1 |
| Ours | **85.9 ± 0.6** | **74.1 ± 0.6** | **87.0 ± 0.3** | **84.8 ± 0.2** | **69.5 ± 0.1** | 93.7 ± 0.2 | **90.8 ± 0.2** |

cluster gather more tightly, which alleviates local cohesion and global sparseness.

The overall loss function $\mathcal{L}$ composed of prototypical loss and reconstruction loss is defined as

$$\mathcal{L} = \mathcal{L}_{pro} + \alpha \mathcal{L}_{rec}, \tag{10}$$

where $\alpha$ is the penalty coefficient. The overall loss function expects that the representations reserve more critical information to restore the raw data, and the cluster distributions in the representation space realize intra-cluster compactness and inter-cluster separateness at the same time.

**Theorem 3.3.** *The GAE with the fuzzy boundary achieves a higher generalization than a vanilla one.*

We provide detailed proof to explain how the fuzzy boundary enhances the model generalization in **Appendix A.2**.

## 4. Experiments

In this section, seven public graph datasets are used for testing the model performance, which ranges from citation networks to co-purchase networks. To show the effectiveness of our model, we compare it with numerous advanced self-supervised graph models proposed recently, i.e., **DeepWalk** (Perozzi et al., 2014), **DGI** (Velickovic et al., 2019), **GMI** (Peng et al., 2020), **GRACE** (Zhu et al., 2020), **GCA** (Zhu et al., 2021), **COAST** (Zhang et al., 2022), **SUGRL** (Mo et al., 2022), **AFGRL** (Lee et al., 2022), **iGCL** (Li et al., 2023a), **NCLA** (Shen et al., 2023), **GREET** (Liu et al., 2023), **HomoGCL** (Li et al., 2023b), **PiGCL** (He et al., 2024b), and **SGRL** (He et al., 2024a). We employ two common downstream tasks, namely node classification

and node clustering to demonstrate the superiority of our model. All experiments are implemented in PyTorch and conducted on a server with RTX 3090.

### 4.1. Node Classification

In the node classification task, we encode the raw graph to obtain corresponding node representations and use a linear classifier to examine the quality of representations. We repeat the experiments 10 times to obtain stable classification results, and mean accuracies with standard deviations are listed in Table 1.

From the results, we find that the proposed model is competitive or surpasses other advanced models across all the datasets. Particularly, our model shows 0.6%, 0.8%, and 0.7% improvements on Citeseer dataset, Ogbn-Arxiv dataset, and Amazon Computers dataset, respectively. Besides, we have the following observations:

(1) Compactness and separateness affect the model performance jointly. Several GCL models without dispersing negatives explicitly, such as AFGRL and iGCL, concentrate on aligning positives to realize compactness. However, they exhibit inferior performance when compared with other models on most datasets, which means separateness also plays a significant role in representation learning.

(2) Global structural distributions with high intra-class compactness and inter-class separateness are easily delimited by our model. Existing GCL models realize compactness and separateness by closing positives and dispersing negatives, but these properties not well reflected by the cluster distribution are likely to limit the model performance. As the

*Table 2.* Results in node clustering. The best results are highlighted in bold, and the second-best results are highlighted with underline.

| Datasets | Metrics | iGCL | GREET | HomoGCL | SGRL | PiGCL | Ours |
|---|---|---|---|---|---|---|---|
| Cora | Acc | 68.56 ± 2.16 | 71.83 ± 3.40 | 70.68 ± 3.85 | 64.76 ± 0.04 | 64.68 ± 2.64 | **72.85 ± 0.02** |
| | MaF1 | 62.65 ± 3.31 | 64.77 ± 5.31 | 66.10 ± 5.87 | 63.82 ± 0.02 | 61.45 ± 3.55 | **71.50 ± 0.01** |
| | NMI | 51.28 ± 1.34 | 55.41 ± 1.16 | **56.44 ± 1.77** | 48.28 ± 0.04 | 51.87 ± 1.78 | 54.71 ± 0.01 |
| | ARI | 45.39 ± 2.05 | **50.42 ± 4.32** | 49.97 ± 2.85 | 39.83 ± 0.07 | 42.84 ± 2.68 | 50.11 ± 0.01 |
| Citeseer | Acc | 66.10 ± 0.58 | 68.78 ± 0.18 | 65.65 ± 0.83 | 64.48 ± 0.05 | 65.45 ± 1.32 | **70.25 ± 0.09** |
| | MaF1 | 61.61 ± 3.67 | 63.57 ± 0.08 | 61.65 ± 0.73 | 61.97 ± 0.05 | 61.10 ± 1.00 | **65.10 ± 0.05** |
| | NMI | 39.42 ± 0.41 | 43.62 ± 0.21 | 39.50 ± 0.58 | 39.67 ± 0.10 | 39.71 ± 0.69 | **44.99 ± 0.03** |
| | ARI | 40.03 ± 0.45 | 44.57 ± 0.31 | 39.73 ± 0.08 | 43.65 ± 0.24 | 39.97 ± 0.94 | **46.68 ± 0.09** |
| Amazon Photo | Acc | 67.72 ± 3.67 | 58.96 ± 4.15 | 74.30 ± 1.69 | 53.39 ± 1.47 | 58.31 ± 3.48 | **76.71 ± 0.06** |
| | MaF1 | 65.88 ± 2.58 | 55.51 ± 1.74 | 69.03 ± 0.64 | 49.49 ± 2.13 | 55.49 ± 2.86 | **75.85 ± 0.82** |
| | NMI | 59.87 ± 2.50 | 50.35 ± 0.60 | 64.21 ± 2.11 | 46.81 ± 1.51 | 47.48 ± 2.02 | **67.72 ± 1.22** |
| | ARI | 45.65 ± 3.40 | 38.76 ± 2.11 | 54.95 ± 3.26 | 27.44 ± 2.51 | 30.51 ± 2.42 | **57.84 ± 1.11** |
| Amazon Computers | Acc | 53.49 ± 2.47 | 51.27 ± 0.15 | 46.75 ± 4.87 | 48.75 ± 0.96 | 41.88 ± 0.68 | **55.04 ± 0.78** |
| | MaF1 | 48.35 ± 3.03 | 39.19 ± 0.36 | 33.10 ± 4.73 | 46.05 ± 0.64 | 33.75 ± 0.89 | **48.79 ± 0.61** |
| | NMI | 49.87 ± 1.30 | 46.66 ± 0.11 | 42.53 ± 2.61 | 45.35 ± 1.27 | 40.94 ± 0.39 | **56.29 ± 0.62** |
| | ARI | 29.75 ± 1.70 | 32.22 ± 0.22 | 26.91 ± 2.59 | 31.11 ± 1.27 | 28.01 ± 0.42 | **36.38 ± 0.78** |

classifier is a simple linear classifier, its performance heavily depends on the qualities of global structural distributions.

In summary, the success of our model contributes to overcoming the two shortcomings, namely local cohesion and global sparseness, which form the ideal global structural distributions to facilitate the downstream tasks.

### 4.2. Node Clustering

The clustering task is also a common task to evaluate the model performance (Xu et al., 2022; 2024b), where we apply the K-means clustering algorithm to examine the encoded node representations. We repeat the experiments 20 times to harvest reliable outcomes, and adopt the commonly-used accuracy (Acc), normalized mutual information (NMI), adjusted rand index (ARI), and macro-F1 (MaF1) metrics to evaluate the clustering performance. Table 2 displays the result comparisons between our proposed model and other advanced graph models.

Similar to node classification, the proposed model almost receives the best outcomes across four datasets, and it exhibits wonderful Acc and MaF1 particularly. It illustrates that produced node representations of our model are clustered more accurately, and implies that the intra-cluster compactness of the model is indeed effective, namely the samples in one cluster are more likely to share the same label. Besides, the proposed model behaves well in both classification and clustering tasks, which implies that the proposed model is able to generate satisfactory cluster distributions in the embedding space.

### 4.3. Quantitative Analysis

In this experiment, we propose two specific metrics, "Mean Groups" and "Total Isolated Nodes" for evaluating the model's ability to address local cohesion and global sparseness. Additionally, we present the "Group List" and the "List of Isolated Nodes" to help understand these two metrics. The specific explanations and definitions of these metrics are provided in the **Appendix B.3**. **"Group List" and "Mean Groups" reflect the local cohesion**, where "Group List" details the number of independent groups in each class cluster and "Mean Groups" calculates the mean of groups from all class clusters. *The value of "Mean Groups" approaches 1 is more expected.* **"List of Isolated Nodes", "Total Isolated Nodes", and "Mean Groups" reflect global sparseness jointly**, where "List of Isolated Nodes" details the number of isolated nodes for each class cluster, and "Total Isolated Nodes" records the total number of the isolated nodes for each dataset. *The value of "Total Isolated Nodes" approaches 0 is more expected.*

For the experiment setting, we apply DBSCAN (Ester et al., 1996) on the produced node representations to collect the number of independent groups and isolated nodes for each class, and the results are displayed in Table 3. "$Eps$" is a hyperparameter of DBSCAN that limits the maximal distances between the representations in a detected local group. Here, we execute the DBSCAN algorithm with $Eps = 0.5$ and $Eps = 0.7$. In general, when "$Eps$" becomes greater, DBSCAN gets more expected outcomes, namely smaller "Mean Groups" and "Total Isolated Nodes".

The results in Table 3 strongly demonstrate the existence

*Table 3.* Quantitative analysis to evaluate global sparseness and local cohesion. The minimal Mean Groups (MGs) and Total Isolated Nodes (TINs) are highlighted in bold.

| Datasets | Eps | Models | Group List | MGs ↓ | List of Isolated Nodes | TINs ↓ |
|---|---|---|---|---|---|---|
| Citeseer | 0.5 | iGCL | [14, 42, 43, 54, 53, 53] | 43.2 | [204, 403, 439, 421, 358, 278] | 2,103 |
| | | GREET | [8, 9, 12, 21, 15, 12] | 12.8 | [223, 540, 603, 592, 517, 440] | 2,915 |
| | | HomoGCL | [16, 49, 25, 49, 41, 42] | 37.0 | [187, 259, 247, 271, 167, 188] | 1,319 |
| | | SGRL | [13, 21, 8, 14, 3, 14] | 12.2 | [123, 141, 112, 149, 62, 89] | 676 |
| | | PiGCL | [8, 5, 9, 8, 2, 9] | 6.8 | [90, 72, 74, 89, 53, 66] | 444 |
| | | Ours | [5, 5, 3, 3, 3, 4] | **3.8** | [24, 14, 24, 14, 11, 23] | **110** |
| | 0.7 | iGCL | [17, 52, 33, 47, 45, 41] | 39.2 | [183, 252, 258, 258, 167, 164] | 1,282 |
| | | GREET | [17, 37, 21, 28, 28, 25] | 26.0 | [179, 397, 473, 437, 341, 316] | 2,143 |
| | | HomoGCL | [20, 34, 19, 34, 18, 34] | 26.5 | [157, 178, 170, 208, 100, 145] | 958 |
| | | SGRL | [6, 1, 1, 7, 1, 4] | 3.3 | [40, 38, 35, 46, 20, 29] | 208 |
| | | PiGCL | [7, 2, 5, 4, 1, 4] | 3.8 | [66, 65, 60, 88, 35, 56] | 370 |
| | | Ours | [2, 1, 1, 2, 1, 3] | **1.7** | [17, 15, 18, 9, 7, 12] | **78** |
| Amazon Photo | 0.5 | iGCL | [1, 17, 7, 4, 3, 4, 7, 1] | 5.5 | [17, 199, 29, 61, 30, 38, 74, 23] | 471 |
| | | GREET | [4, 18, 6, 10, 8, 8, 18, 3] | 9.4 | [172, 661, 213, 383, 526, 296, 1076, 253] | 3,580 |
| | | HomoGCL | [3, 16, 2, 5, 35, 1, 1, 6, 31, 2] | 10.2 | [39, 251, 21, 67, 445, 18, 13, 55, 225, 33] | 1,167 |
| | | SGRL | [7, 27, 14, 15, 12, 15, 16, 12] | 14.8 | [58, 750, 106, 400, 258, 199, 258, 195] | 2,224 |
| | | PiGCL | [1, 5, 4, 1, 2, 3, 5, 1] | 2.8 | [12, 102, 20, 35, 17, 30, 56, 3] | 275 |
| | | Ours | [1, 3, 2, 2, 1, 1, 2, 2] | **1.8** | [6, 15, 10, 4, 15, 5, 10, 3] | **68** |
| | 0.7 | iGCL | [1, 8, 4, 1, 2, 3, 4, 1] | 3.0 | [8, 108, 21, 38, 18, 29, 52, 4] | 278 |
| | | GREET | [3, 6, 4, 6, 6, 1, 12, 2] | 5.0 | [59, 282, 97, 183, 232, 99, 472, 142] | 1,566 |
| | | HomoGCL | [1, 10, 6, 4, 6, 3, 9, 2] | 5.1 | [17, 144, 37, 67, 45, 41, 85, 21] | 457 |
| | | SGRL | [2, 22, 4, 8, 4, 5, 6, 2] | 6.6 | [22, 338, 36, 100, 49, 56, 79, 47] | 727 |
| | | PiGCL | [1, 3, 3, 1, 2, 1, 2, 1] | 1.8 | [5, 67, 17, 26, 10, 23, 42, 1] | 191 |
| | | Ours | [1, 1, 2, 1, 1, 1, 1, 1] | **1.1** | [5, 6, 2, 3, 1, 5, 4, 1] | **27** |
| Amazon Computers | 0.5 | iGCL | [2, 10, 1, 6, 25, 1, 1, 7, 22, 2] | 7.7 | [14, 161, 24, 50, 329, 16, 14, 43, 165, 30] | 846 |
| | | GREET | [5, 21, 3, 5, 66, 1, 3, 8, 26, 3] | 14.1 | [310, 1024, 263, 155, 2157, 25, 135, 205, 977, 64] | 5,315 |
| | | HomoGCL | [3, 16, 2, 5, 35, 1, 1, 6, 31, 2] | 10.2 | [39, 251, 21, 67, 445, 18, 13, 55, 225, 33] | 1,167 |
| | | SGRL | [1, 13, 4, 4, 16, 1, 3, 5, 18, 5] | 7.0 | [95, 480, 142, 96, 494, 20, 50, 189, 429, 56] | 2,051 |
| | | PiGCL | [3, 5, 2, 3, 15, 1, 1, 3, 8, 3] | 4.4 | [15, 147, 22, 39, 293, 13, 6, 44, 132, 16] | 727 |
| | | Ours | [3, 4, 2, 1, 9, 3, 1, 4, 6, 3] | **3.6** | [20, 89, 18, 18, 119, 6, 10, 27, 103, 18] | **428** |
| | 0.7 | iGCL | [2, 6, 3, 2, 19, 1, 1, 6, 14, 3] | 5.7 | [16, 129, 26, 29, 267, 12, 10, 53, 131, 16] | 689 |
| | | GREET | [4, 10, 2, 5, 28, 2, 1, 1, 16, 2] | 7.1 | [46, 237, 9, 42, 443, 12, 23, 33, 255, 24] | 1,124 |
| | | HomoGCL | [3, 6, 1, 3, 15, 1, 1, 5, 13, 2] | 5.0 | [7, 114, 11, 33, 255, 16, 12, 23, 108, 20] | 599 |
| | | SGRL | [1, 3, 1, 1, 8, 1, 1, 5, 5, 2] | 2.8 | [25, 113, 18, 22, 234, 10, 7, 53, 123, 21] | 626 |
| | | PiGCL | [2, 3, 2, 2, 8, 1, 1, 2, 6, 2] | 2.9 | [3, 75, 5, 20, 185, 11, 4, 19, 64, 12] | 398 |
| | | Ours | [2, 2, 1, 2, 2, 1, 1, 2, 2, 1] | **1.6** | [6, 18, 6, 4, 21, 2, 2, 7, 9, 7] | **82** |

of global sparseness and local cohesion in GCL, where the groups and isolated nodes in the clusters are much higher than ideal conditions, namely "Mean Groups" is 1 and "Total Isolated Nodes" is 0. We also observe that the proposed model has the most expected "Mean Groups" and "Total Isolated Nodes" when compared with other advanced GCL models under all the settings, which means the proposed fuzzy boundary effectively alleviates the two shortcomings.

### 4.4. Benefits from Compactness

In this experiment, we project the raw data into a 3-dimension hypersphere for visualization, and the representations after pre-training and fine-tuning are shown in Figure 4(a) and (b) respectively. Since the raw high-dimension data are condensed into a 3-dimension space, it leads to abundant

information loss, and the result after pre-training is terrible. Even so, under the guidance of fuzzy boundaries, the distribution of each cluster becomes clearer, and the samples of each cluster get more compact. Since fuzzy boundaries are the outside of the real clusters, narrowing the distances between prototypes and their corresponding fuzzy boundaries effectively tighten the real clusters.

## 5. Conclusion

In this paper, we first attend to two shortcomings of existing GCL models, namely global sparseness and local cohesion, which lead to inferior global structural distribution. Such a distribution imperils the representation generalization and expressiveness, and is adverse to downstream tasks. Considering this, we design a novel fuzzy boundary that

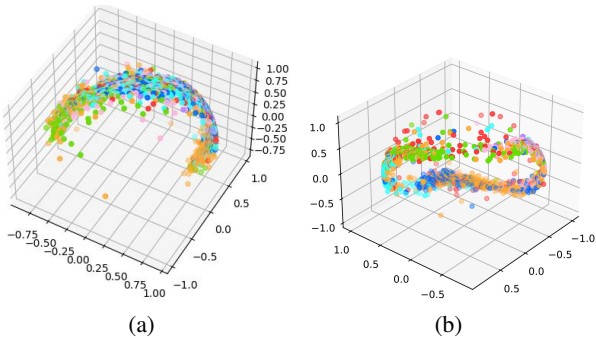

(a)         (b)

*Figure 4.* Representation distributions after pre-training and fine-tuning on the Cora dataset. (a) is the representation distribution after pre-training and it is mixed, but becomes distinct with the fuzzy boundary after fine-tuning as (b).

includes fuzzy boundary construction and fuzzy boundary contraction phases. By implementing the fuzzy boundary on a pre-trained GAE, the cluster distributions achieve premium global structural properties, namely high intra-cluster compactness and inter-cluster separateness. The extensive experiments on two downstream tasks and the quantitative analysis comprehensively illustrate the merits of the fuzzy boundary to address these shortcomings.

## Acknowledgements

This work was supported by the National Key Research and Development Program of China (No. 2023YFB3107100), the National Natural Science Foundation of China (No. 62306020), the Young Elite Scientist Sponsorship Program by BAST (No. BYESS2024199), Beijing Natural Science Foundation (No. L244009), the National Social Science Fund of China (No. 23VRC094), and the Major Program of the National Social Science Foundation of China (No. 22&ZD147). The first author is funded by the China Scholarship Council (CSC) from the Ministry of Education, China.

## Impact Statement

This paper presents work whose goal is to advance the field of Machine Learning. There are many potential societal consequences of our work, none which we feel must be specifically highlighted here.

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

# A. Theoretical Proof

## A.1. Theorem 3.2

*Proof:* Since we use the centralized representations for constructing the fuzzy boundary, the overall variances within each cluster are formed as

$$
\begin{aligned}
\min \mathcal{J} &= \min \sum_{c=1}^{C_p} \sum_{i \in C_c} \|\bar{\mathbf{z}}_i\|_2^2 \\
&= \min \sum_{c=1}^{C_p} \sum_{i \in C_c} \|(\mathbf{2} - \tilde{\mathbf{s}}_i) \odot \mathbf{z}_i\|_2^2 \\
&= \min \sum_{i=1}^{N} \|(\mathbf{2} - \tilde{\mathbf{s}}_i) \odot \mathbf{z}_i\|_2^2,
\end{aligned}
\tag{11}
$$

where $C_p$ is the number of the clusters, $C_c$ is the index set including the node indices of the $c$-th cluster. For the $j$-th element of $\tilde{\mathbf{s}}_i$, We have following observation:

$$
\frac{\partial \mathcal{J}}{\partial \tilde{\mathbf{s}}_{i,j}} = -2\left(2 - \tilde{\mathbf{s}}_{i,j}\right) \cdot \mathbf{z}_{i,j}^2 \leq 0.
\tag{12}
$$

Therefore, $\mathcal{J}$ is a monotonically decreasing function of $\tilde{\mathbf{s}}_{i,j}$. When applying *max-pooling* as the $Union$ operator, we have

$$
\begin{aligned}
&\|(\mathbf{2} - \{\tilde{\mathbf{s}}_{max}\}_i) \odot \mathbf{z}_i\|_2^2 \\
&= \sum_j \left[(\mathbf{2} - \{\tilde{\mathbf{s}}_{max}\}_{i,j}) \cdot \mathbf{z}_{i,j}\right]^2 \\
&\leq \sum_j \left[(\mathbf{2} - \{\tilde{\mathbf{s}}_{other}\}_{i,j}) \cdot \mathbf{z}_{i,j}\right]^2 \\
&\leq \sum_j \left[(\mathbf{2} - \{\tilde{\mathbf{s}}_{min}\}_{i,j}) \cdot \mathbf{z}_{i,j}\right]^2.
\end{aligned}
\tag{13}
$$

where $\{\tilde{\mathbf{s}}_{min}\}_i$ is the membership degree vector of node $i$, which is filled with minimal membership degrees, and its element is $\{\tilde{\mathbf{s}}_{min}\}_{i,j} = \min_k\{\mathbf{s}_{i,j}^k\}$. Similarly, we have $\{\tilde{\mathbf{s}}_{max}\}_{i,j} = \max_k\{\mathbf{s}_{i,j}^k\}$. $\{\tilde{\mathbf{s}}_{other}\}_i$ denotes other simple $Union$ operators. Then we have the following conclusion as

$$
\begin{aligned}
&\sum_{i=1}^{N} \|(\mathbf{2} - \{\tilde{\mathbf{s}}_{max}\}_i) \odot \mathbf{z}_i\|_2^2 \\
&\leq \sum_{i=1}^{N} \|(\mathbf{2} - \{\tilde{\mathbf{s}}_{other}\}_i) \odot \mathbf{z}_i\|_2^2 \\
&\leq \sum_{i=1}^{N} \|(\mathbf{2} - \{\tilde{\mathbf{s}}_{min}\}_i) \odot \mathbf{z}_i\|_2^2.
\end{aligned}
\tag{14}
$$

Finally, we get the minimum of $\mathcal{J}$ is $\sum_{i=1}^{N} \|(\mathbf{2} - \{\tilde{\mathbf{s}}_{max}\}_i) \odot \mathbf{z}_i\|_2^2$, when the $Union$ operator is max-pooling as $\tilde{\mathbf{s}}_{i,j} = Union\{\mathbf{s}_{i,j}^k\} = \max_k\{\mathbf{s}_{i,j}^k\}$.

## A.2. Theorem 3.3

We first introduce **Definition A.1** and **Lemma A.2**, which are the core of the theorem proof.

**Definition A.1.** Complexity Measure based on the Davies-Bouldin Index. The Davies-Bouldin Index is calculated by the ratio of intra-cluster compactness to inter-cluster separateness (Natekar & Sharma, 2020), and it stands for the consistency within a cluster and the difference among clusters. Mathematically, it is defined by

$$
B = \frac{1}{C_p} \sum_{m=0}^{C_p-1} \max_{m \neq n} \frac{Q_m + Q_n}{U_{m,n}},
\tag{15}
$$

where

$$Q_m = \left( \frac{1}{N_m} \sum_i \|\mathbf{Z}_i^m - \mu_m\|_2^2 \right)^{\frac{1}{2}}, \quad \text{for } m = 1, \cdots, C_p, \tag{16}$$

$$U_{m,n} = \|\mu_m - \mu_n\|_2, \quad \text{for } m, n = 1, \cdots, C_p.$$

$m$ and $n$ are the $m$-th cluster and $n$-th cluster, $C_p$ is the number of clusters, $\mathbf{Z}_i^m$ denotes representation of the $i$-th sample in the $m$-th cluster, $\mu_m$ is the prototype of the $m$-th cluster, $Q_m$ measures the intra-cluster compactness of cluster $m$, and $U_{m,n}$ measures the inter-cluster separateness between cluster $m$ and cluster $n$ (Davies & Bouldin, 1979; Natekar & Sharma, 2020).

**Lemma A.2.** *The fuzzy boundary leads to a lower bound of the complexity measure $B$, which is in line with a higher generalization bound of the model.*

*Proof:* We follow the proof idea of (Huang et al., 2024). For simplicity, we consider the condition with two clusters. In this case, the complexity measure $B$ is $\frac{Q_0 + Q_1}{U_{0,1}}$. We assume the probability that a node belongs to cluster $i$ is $P_i$. For calculating $B$, we firstly compute the cluster prototypes $\mu_0$ and $\mu_1$ as

$$\mu_0 = \mathbb{E}[\mathbf{Z}_i^0] = \mathbb{E}\left[ \mathbf{W}\left( \mathbf{X}_i + \sum_{j \in \mathcal{N}(v_i)} \frac{1}{d_i}\mathbf{X}_j \right) \right] = \mathbf{W}\left( P_0 \cdot \mu_{\mathbf{X}_0} + (1 - P_0) \cdot \mu_{\mathbf{X}_1} \right), \tag{17}$$

$$\mu_1 = \mathbf{W}\left( P_1 \cdot \mu_{\mathbf{X}_1} + (1 - P_1) \cdot \mu_{\mathbf{X}_0} \right).$$

where $\mathbf{W}$ is the parameter matrix of the encoder, $\mu_{\mathbf{X}_i}$ is the prototype of cluster $i$ in the original feature space, $\mathcal{N}(v_j)$ stands for the neighbors of node $v_j$.

Therefore, $Q_0$ and $Q_1$ are calculated as

$$\begin{aligned}
Q_0 &= \sqrt{Q_0^2} \approx \sqrt{\mathbb{E}\left[\|\mathbf{Z}_i^0 - \mu_0\|_2^2\right]} \\
&= \sqrt{\mathbb{E}\left[P_0^2(\mathbf{X}_0 - \mu_{\mathbf{X}_0})^T\mathbf{W}^T\mathbf{W}(\mathbf{X}_0 - \mu_{\mathbf{X}_0})\right] + \mathbb{E}\left[(1-P_0)^2(\mathbf{X}_1 - \mu_{\mathbf{X}_1})^T\mathbf{W}^T\mathbf{W}(\mathbf{X}_1 - \mu_{\mathbf{X}_1})\right]} \\
&= \sqrt{P_0^2\mathbb{E}\left[\|\mathbf{W}(\mathbf{X}_0 - \mu_{\mathbf{X}_0})\|_2^2\right] + (1-P_0)^2\mathbb{E}\left[\|\mathbf{W}(\mathbf{X}_1 - \mu_{\mathbf{X}_1})\|_2^2\right]}, \\
Q_1 &= \sqrt{P_1^2\mathbb{E}\left[\|\mathbf{W}(\mathbf{X}_1 - \mu_{\mathbf{X}_1})\|_2^2\right] + (1-P_1)^2\mathbb{E}\left[\|\mathbf{W}(\mathbf{X}_0 - \mu_{\mathbf{X}_0})\|_2^2\right]}.
\end{aligned} \tag{18}$$

Let $\sigma_i^2 = \mathbb{E}\left[\|\mathbf{W}(\mathbf{X}_i - \mu_{\mathbf{X}_i})\|_2^2\right]$, then we have

$$\begin{aligned}
Q_0 &= \sqrt{P_0^2\sigma_0^2 + (1-P_0)^2\sigma_1^2}, \\
Q_1 &= \sqrt{P_1^2\sigma_1^2 + (1-P_1)^2\sigma_0^2}.
\end{aligned} \tag{19}$$

And let $Q_0' = P_0^2\sigma_0^2 + (1-P_0)^2\sigma_1^2$, we can calculate $\frac{\partial Q_0'}{\partial P_0} = 0$ and get $P_0^* = \frac{\sigma_1^2}{\sigma_0^2 + \sigma_1^2}$. Substitute $P_0^*$ into $Q_0$, we can obtain

$$Q_0 \geq \frac{\sqrt{2}\sigma_0\sigma_1}{\sqrt{\sigma_0^2 + \sigma_1^2}}. \tag{20}$$

Similarly, we also obtain

$$Q_1 \geq \frac{\sqrt{2}\sigma_0\sigma_1}{\sqrt{\sigma_0^2 + \sigma_1^2}}. \tag{21}$$

Let $\widetilde{Q} = \frac{\sigma_0\sigma_1}{\sqrt{\sigma_0^2 + \sigma_1^2}}$, we find that

$$\frac{\partial\widetilde{Q}}{\partial\sigma_0} = \frac{\sigma_1^3}{\sqrt[3]{(\sigma_0^2 + \sigma_1^2)^2}} \geq 0, \frac{\partial\widetilde{Q}}{\partial\sigma_1} = \frac{\sigma_0^3}{\sqrt[3]{(\sigma_0^2 + \sigma_1^2)^2}} \geq 0. \tag{22}$$

Therefore, $Q_0$ and $Q_1$ increase when $\sigma_0$ and $\sigma_1$ increase respectively, and the complexity measure $B$ has the lower bound as

$$B = \frac{Q_0 + Q_1}{U_{0,1}} \geq \frac{2\sqrt{2}\sigma_0\sigma_1}{\sqrt{\sigma_0^2 + \sigma_1^2}\|\mu_0 - \mu_1\|}. \tag{23}$$

Let $(\sigma_i')^2 = \mathbb{E}\left[\|\mathbf{W}(\mathbf{X}_i - \mu_{\mathbf{X}_i}')\|_2^2\right]$ and $\mu_i'$ be the variance and prototype of real cluster when applying fuzzy boundary respectively. Since a vanilla model applies the fuzzy boundary would increase the inter-cluster separateness and reduce the intra-cluster compactness of the embedding distribution, we can get $\sigma_i' \leq \sigma_i$ and $\|\mu_0 - \mu_1\| \leq \|\mu_0' - \mu_1'\|$. Then we finally obtain the following equation as

$$B \geq \frac{2\sqrt{2}\sigma_0\sigma_1}{\sqrt{\sigma_0^2 + \sigma_1^2}\|\mu_0 - \mu_1\|} \geq \frac{2\sqrt{2}\sigma_0'\sigma_1'}{\sqrt{(\sigma_0')^2 + (\sigma_1')^2}\|\mu_0' - \mu_1'\|}. \tag{24}$$

which means the fuzzy boundary could reduce the lower bound of $B$. Following the conclusion of (Natekar & Sharma, 2020; Huang et al., 2024), a lower bound of the complexity measure corresponds to an upper bound of the generalization ability.

# B. Experiments

## B.1. Datasets

We conduct experiments on seven well-known public datasets, and the detailed statistics are summarized in Table 4. **Cora**, **CiteSeer**, **Pubmed**, and **DBLP** are the famous citation networks. Every node in these datasets is a paper, and the link between two nodes means the citation relationship. Each dimension of the node attribute vector refers to a keyword of the paper. **Amazon Photo** and **Amazon Computers** are two co-purchase graphs where the nodes are products, and the edges between the two products show they are frequently bought together. Every node feature is constructed by a bag-of-words representation. **Ogbn-arxiv** is extracted from the Microsoft Academic Graph (Hu et al., 2020). It is a citation network constructed by Computer Science arXiv papers, whose nodes denote papers and edges are citation relations. Each node is obtained by averaging the skip-gram word embeddings from the title and abstract.

*Table 4.* Statistics of different graph datasets.

| Dataset | Cora | Citeseer | Pubmed | DBLP | Amazon Photo | Amazon Computers | Ogbn-arxiv |
|---|---|---|---|---|---|---|---|
| Nodes | 2,708 | 3,327 | 19,717 | 17,716 | 7,650 | 13,752 | 169,343 |
| Edges | 5,429 | 4,732 | 44,338 | 52,867 | 119,081 | 491,722 | 1,166,243 |
| Features | 1,433 | 3,703 | 500 | 1,639 | 745 | 767 | 128 |
| Classes | 7 | 6 | 3 | 4 | 8 | 10 | 40 |

## B.2. Hyperparameter Setting

The hyperparameter setting is listed in Table 5. Pre_Epochs, Pre_Lr, and Pre_Wd are the training epochs, the learning rate, and the weight decay applied in the pre-trained processes. Dim_Hid, Dim_Emb, and Dim_Proj are the hidden dimension of the GNN encoder, the output dimension of the GNN encoder, and the hidden dimension of the projection head, respectively. Act is the used activation function for all components of the proposed model. FT_Lr, FT_Epochs, and FT_Wd are the learning rate, the training epochs, and the weight decay applied in the fine-tuning processes. $T$ is used for updating crisp prototypes. $K$ is the number of the membership functions for each element of a centralized feature vector $\mathbf{z}$, $\tau$ is set to weigh the significance of the distances between the fuzzy boundary and the prototypes, $C_p$ is the number of the clusters for each dataset, $\alpha$ is the penalty coefficient.

## B.3. Details on Quantitative Analysis

The quantitative analysis measures the model ability to address the local cohesion and global sparseness, where we propose two metrics "Mean Groups" and "Total Isolated Nodes" to measure the local cohesion and global sparseness for the quantitative analysis, and present "Group List" and "List of Isolated Nodes" to assist in understanding the two metrics. Here, we provide some definitions and explanations for them:

*Table 5.* Hyperparameter setting on different datasets.

| Dataset | Cora | Citeseer | Pubmed | DBLP | Amazon Photo | Amazon Computers | Ogbn-arxiv |
|---|---|---|---|---|---|---|---|
| Pre_Epochs | 900 | 900 | 1900 | 800 | 500 | 600 | 30 |
| Pre_Lr | 5e-5 | 5e-5 | 5e-5 | 5e-5 | 5e-5 | 5e-5 | 3e-5 |
| Pre_Wd | 1e-5 | 1e-5 | 5e-6 | 5e-6 | 5e-6 | 1.5e-4 | 1e-5 |
| Dim_Hid | 512, 512 | 512, 512 | 1024 | 512 | 512 | 512 | 256, 256 |
| Dim_Emb | 256 | 256 | 1024 | 256 | 256 | 256 | 256 |
| Act | ELU | ELU | ELU | ELU | ELU | ELU | ELU |
| Dim_Proj | 512 | 512 | 2048 | 512 | 512 | 512 | 512 |
| FT_Lr | 5e-5 | 5e-5 | 5e-5 | 8e-5 | 7e-5 | 8e-5 | 1e-5 |
| FT_Wd | 2e-5 | 6e-6 | 1e-5 | 5e-6 | 1e-4 | 1e-4 | 1e-4 |
| FT_Epochs | 700 | 300 | 4500 | 5000 | 6500 | 6000 | 30 |
| $T$ | 40 | 20 | 40 | 40 | 40 | 40 | 40 |
| Dropout ratio | 0.5 | 0.5 | 0.5 | 0.5 | 0.5 | 0.5 | 0.5 |
| $K$ | 5 | 5 | 5 | 5 | 5 | 5 | 5 |
| $\tau$ | 3.5 | 3.2 | 7.0 | 4.5 | 3.2 | 3.5 | 6.0 |
| $C_p$ | 30 | 30 | 550 | 250 | 100 | 150 | 1000 |
| $\alpha$ | 0.9 | 0.0 | 0.3 | 1.0 | 0.9 | 1.1 | 0.8 |

(1) "Group List" and "Mean Groups" reflect the local cohesion. "Group List" is a list containing the number of groups detected within each class cluster, with each element in the list corresponding to the number of groups in a given cluster. A group is a set of nodes that are close to each other in the embedding space, with the maximal distance between each node pair smaller than pre-defined "Eps", where "$Eps$" is a hyperparameter of DBSCAN to limit the maximal distance between the representations in a detected local group. "Mean Groups" calculates the mean number of groups from all the classes. The calculation of "Mean Groups" is given by

$$Mean\ Groups = \frac{1}{C}\sum_{i=1}^{C} G_i, \tag{25}$$

where $C$ is the number of classes, $G_i$ is the number of groups for the $i$-th class cluster. For an ideal cluster distribution, all the samples with the same label are supposed to be gathered in one cluster, so each element in the "Group List" is expected to approach 1, the same as "Mean Groups".

(2) "List of Isolated Nodes", "Total Isolated Nodes", and "Mean Groups" reflect global sparseness jointly. "List of Isolated Nodes" details the number of isolated nodes for each class cluster, and the isolated nodes are the nodes whose distances between themselves and any other group are greater than "$Eps$", and these nodes are not assigned to any group and contribute to global sparseness. "Total Isolated Nodes" records the total number of isolated nodes for each dataset. The calculation of "Total Isolated Nodes" is given by

$$Total\ Isolated\ Nodes = \sum_{i=1}^{C} I_i, \tag{26}$$

where $I_i$ is the number of isolated nodes for the $i$-th class cluster. Generally, ideal cluster distributions hardly have isolated nodes. According to the aforementioned explanations, smaller "Mean Groups" and "Total Isolated Nodes" could reflect better clustering performance in general.

### B.4. Hyperparameter Analysis

We analyze two hyperparameters directly related to the loss function, namely the temperature coefficient $\tau$ and penalty coefficient $\alpha$, whose results are exhibited in Figure 5. Not surprisingly, temperature coefficient $\tau$ plays a significant role in the model performance as it weighs the distances between the prototypes and the fuzzy boundaries. Greater $\tau$ makes the distances between prototypes and their corresponding fuzzy boundaries much closer than those between different prototypes.

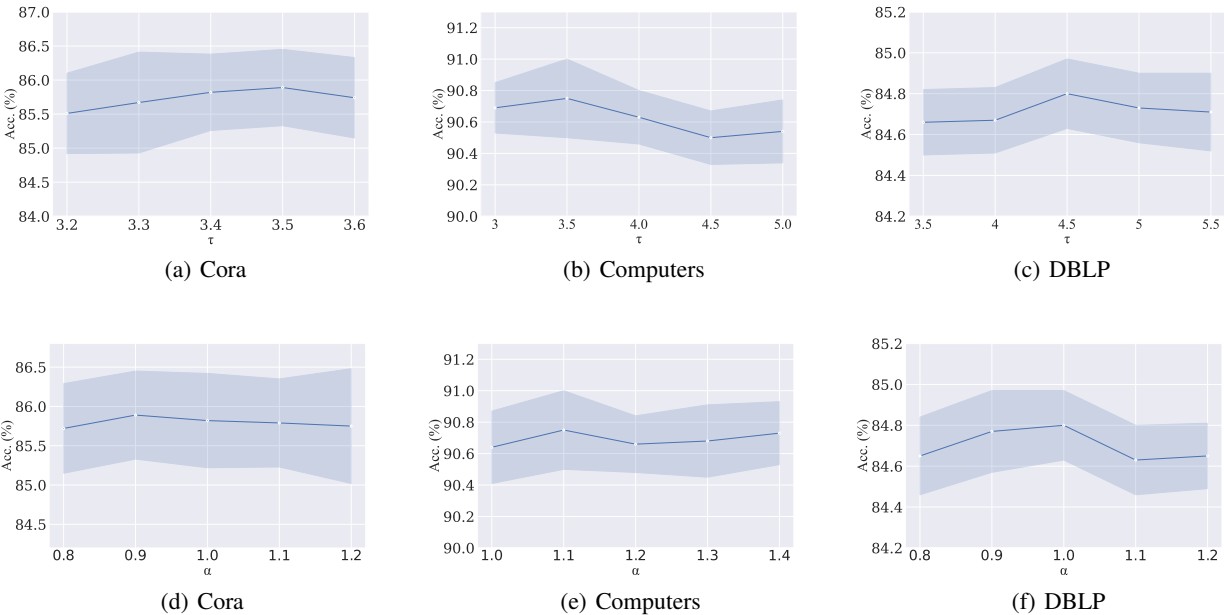

*Figure 5.* Results of hyperparameter sensitivity analysis, $\tau$ affects the model performance significantly and $\alpha$ influences slightly.

In other words, relatively larger $\tau$ balances the intra-cluster compactness and inter-cluster separateness. In contrast to the function of $\tau$, $\alpha$ influences the model performance slightly. Even so, it functions to preserve the critical information as much as possible.

### B.5. Visualization

In this experiment, we draw the t-sne of our model after pre-training and fine-tuning respectively to show how the fuzzy boundary forms the ideal global structural distribution in Figure 6. We find that after pre-training, the cluster distributions display global sparseness. However, after fine-tuning, the global sparseness disappears and the ideal properties of cluster distributions, namely intra-cluster compactness and inter-cluster separateness gradually appear. Besides, by comparing Figure 6(b) with (c), we find that the fuzzy boundaries indeed extend the spectrums of the real clusters. And when we realize the contraction on outer fuzzy boundaries, the inner real clusters gather more tightly. Except for improving intra-cluster compactness, the fuzzy boundaries implicitly enhance inter-cluster separateness. Concretely, the fuzzy clusters, namely the clusters that include both real samples and fuzzy boundary elements are separatable (in Figure 6(b)), and when the fuzzy clusters further distance each other, the distances between the inner real clusters are saliently enlarged (in Figure 6(c)). As a result, the clusters embody high intra-cluster compactness and inter-class separateness merits.

### B.6. Complexity

The complexity of the proposed model mainly includes the encoding process, K-means clustering algorithm, centralization process, calculation of fuzzy boundary, calculation of the reconstruction loss, and calculation of the prototypical contrastive loss. The encoding process demands $\mathcal{O}(LEd)$ complexity, where $L$ is the layer number of the encoder, $E$ is the number of the edges, $d$ is the dimension of the output representation. The K-means clustering algorithm demands $\mathcal{O}(C_pNd)$ complexity, where $C_p$ is the number of prototypes, $N$ is the number of samples. The centralization process, the calculation of fuzzy boundary, and the reconstruction loss all demand $\mathcal{O}(Nd)$ complexity. The prototypical contrastive loss demands $\mathcal{O}(C_pNd^2)$ complexity. Overall, the whole complexity of the proposed model is $\mathcal{O}(LEd + C_pNd + 3Nd + C_pNd^2)$.

### B.7. Ablation Study

Table 6 lists the node classification results in the ablation study, where Pre-training means the results for pre-trained GAE, Fuzzy Boundary corresponds to the model with the fuzzy boundary, $\mathcal{L}_{pro}$ is the contrastive loss, and $\mathcal{L}_{rec}$ is the

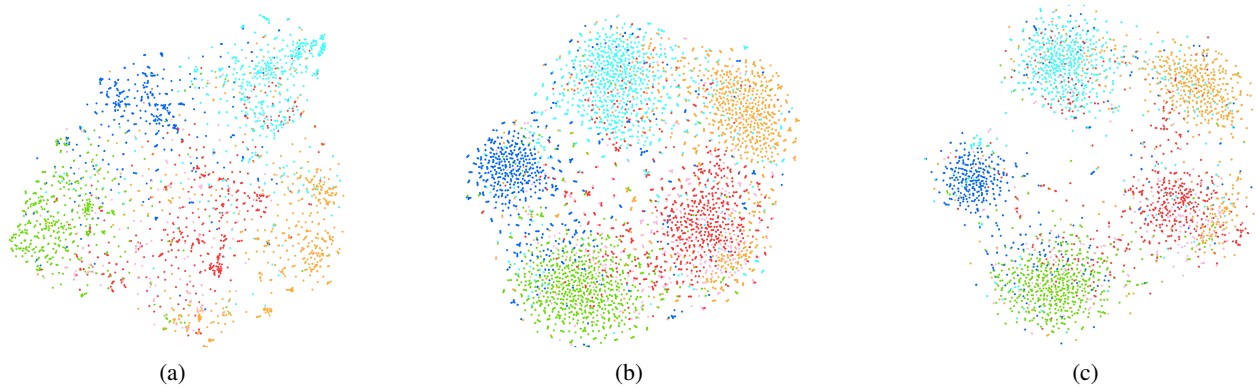

(a)  (b)  (c)

*Figure 6.* t-SNE results after pre-training and fine-tuning on Citeseer dataset, where (a) includes the crisp representations after pre-training, (b) includes both crisp representations and the fuzzy boundary elements after fine-tuning, and (c) is solely composed of the node representations after fine-tuning. Cluster distributions of (b) and (c) show ideal intra-cluster compactness and inter-cluster separateness.

*Table 6.* Results of ablation study on node classification task, which are concerned with the functions of pre-training, fuzzy boundary, prototypical contrastive loss $\mathcal{L}_{pro}$, and reconstruction loss $\mathcal{L}_{rec}$.

| Pre-training | Fuzzy Boundary | $\mathcal{L}_{pro}$ | $\mathcal{L}_{rec}$ | Cora | DBLP | Amazon Photo | Amazon Computers |
|:---:|:---:|:---:|:---:|:---:|:---:|:---:|:---:|
| ✓ | | | | 84.83 ± 0.75 | 83.40 ± 0.35 | 93.21 ± 0.20 | 89.72 ± 0.24 |
| ✓ | | ✓ | | 85.24 ± 0.55 | 84.33 ± 0.16 | 93.43 ± 0.21 | 90.03 ± 0.15 |
| ✓ | ✓ | ✓ | | 85.73 ± 0.62 | 84.67 ± 0.17 | 93.60 ± 0.21 | 90.54 ± 0.20 |
| | ✓ | ✓ | ✓ | 84.86 ± 0.58 | 84.35 ± 0.14 | 93.53 ± 0.19 | 90.46 ± 0.22 |
| ✓ | ✓ | ✓ | ✓ | **85.90 ± 0.56** | **84.80 ± 0.17** | **93.71 ± 0.20** | **90.75 ± 0.20** |

reconstruction loss. Compared with models without fuzzy boundary, the models with fuzzy boundary behaves better, which means the fuzzy boundary significantly enhances model performance. By combing $\mathcal{L}_{pro}$ with $\mathcal{L}_{rec}$, the model performance is further improved.

