# OpenReview forum: "Mitigating Local Cohesion and Global Sparseness in Graph Contrastive Learning with Fuzzy Boundaries"
_ICML.cc/2025/Conference — ICML 2025 poster_

### Official Review · Reviewer_6dir · 2025-02-25

**Overall Recommendation:** 4

**Summary:**

This paper introduces fuzzy boundaries to address ingrained local cohesion and global sparseness in graph contrastive learning (GCL). To address these issues, this paper proposes a novel GCL model that replaces the crisp cluster boundaries with adaptive fuzzy boundaries to adjust the cluster boundaries. Extensive experiments demonstrate that the proposed method consistently outperforms baseline models across multiple downstream tasks.

**Claims And Evidence:**

The claims made in the submission are supported by relatively convincing evidence. The paper first uses the visualization to detail the problem, and the experiment results show that the fuzzy boundary could solve the problem.

**Essential References Not Discussed:**

The references are enough for me to follow the idea, and I have no essential references to supplement.

**Experimental Designs Or Analyses:**

The paper provides sound experimental designs to validate the model performance across multiple datasets and two downstream tasks. The analyses of the experiments are plausible.

**Methods And Evaluation Criteria:**

The proposed model extends the concept of crisp boundaries by introducing fuzzy boundaries, leveraging fuzzy set theory to dilate and then contract these boundaries. In this way, the fuzzy boundaries could tighten all the clusters. Technically, it makes sense for addressing local cohesion and global sparseness.

**Other Comments Or Suggestions:**

Please refer to the weaknesses.

**Other Strengths And Weaknesses:**

This paper introduces fuzzy boundaries to improve intra-cluster compactness and inter-cluster separateness in contrastive learning, and the method alleviates the shortcomings to realize satisfactory global structure distributions. Additionally, the paper validates the proposed method across multiple datasets, demonstrating its effectiveness in improving node classification and clustering tasks. Still, the paper has suffered from the following weaknesses.

Weaknesses

The fuzzy boundary only adopts max-pooling as the Union operator. Even though providing the minimal variances theoretically, it is better to provide more experiments with other Union operators in practice to illustrate that max-pooling is a sensible choice.

**Questions For Authors:**

1.  In the construction of fuzzy boundaries using crisp and fuzzy representations, why does the author choose to use a coordinate difference that is multiplied by one?
2. Why do the authors use the visualization of DGI, GRACE, and BGRL to illustrate the shortcomings? Since these models are not the most advanced graph contrastive models, it may cause a problem whether the shortcomings emerge in all the graph contrastive models.
3. Some results in Quantitative Analysis are confusing. For example, the “MGs” of GREET with “EPS=0.5” and “EPS=0.7” on Citeseer shows high “EPS” will increase “MGs”. However, it is contradictory to the statement that the model gets more expected outcomes when “Eps” becomes greater in Section 4.3.
4. Why aren't all methods compared in the Node Clustering task?
5. The negative correlation methods in ensemble learning aim to ensure that while individual learners predict correctly, they also exhibit greater diversity, which aligns with the goals of this paper. Can the authors analyze similar ideas and methods?
---
Thank you for your thoughtful responses. All of my concerns have been thoroughly addressed. I decided to increase my scores.

**Relation To Broader Scientific Literature:**

This paper has found two shortcomings in graph contrastive learning and proposes the fuzzy boundary to solve them. These shortcomings seem to be alleviated according to the visualization results between the proposed model and prior related research.

**Theoretical Claims:**

I have checked the proof carefully and think the theorem is almost right.

---

> ### Author Rebuttal · Authors · 2025-03-30
>
> **W1.**
>
> We chose Max Pooling as the Comb operator due to its theoretical advantage, however, it is indeed necessary to compare it with other pooling strategies in practice. We have conducted additional experiments on other Comb operators, and the results on node classification are shown in Table 1.
>
> From the results, we could find that the Max Pooling could get the best results across all the datasets since Max Pooling better tightens the samples within each cluster when compared with other general operators. Besides, the samples inside the fuzzy boundary constructed with Max Pooling have the greatest degree belonging to the corresponding cluster, which effectively avoids incurring samples from other clusters.
>
> Table 1. Comparisons among different Comb operators on node classification.
>
> |Comb|Cora|Citeseer|Pubmed|DBLP|Arxiv|Photo|Computers|
> |-|-|-|-|-|-|-|-|
> |Min|85.5±0.6|73.3±0.4|86.3±0.3|83.6±0.3|69.0±0.0|92.8±0.2|89.3±0.3|
> |Mean|85.7±0.5|73.5±0.7|86.6±0.2|84.1±0.3|69.2±0.0|93.1±0.2|90.1±0.2|
> |Max|**85.9±0.6**|**74.1±0.6**|**87.0±0.3**|**84.8±0.2**|**69.5±0.1**|**93.7±0.2**|**90.8±0.2**|
>
> **Q1.**
>
> I guess you may wonder why we choose to use a coordinate difference that is multiplied by one rather than another scaling factor, and I give the following answers. Firstly, the coordinate difference between crisp and fuzzy representations is used to define an acceptable distance (semantic discrepancy) between positives, which is in line with the condition that the scaling factor is one. Secondly, the membership degree is adjusted adaptively during the training process, which can also be viewed as a scaling factor to construct the fuzzy boundaries, so we do not need to set an additional fixed scaling factor.
>
> **Q2.**
>
> We chose DGI, GRACE, and BGRL as representative models because they are the three types most representative of current contrastive learning approaches. While these models are not the most recent graph contrastive models, they represent mainstream and fundamental GCL paradigms: (1) DGI focuses on a global-local contrastive framework. (2) GRACE adopts a node-level contrastive framework. (3) BGRL is a non-negative-sample-based GCL method. Even though these models adopt different learning paradigms, they share similar shortcomings in the representation distribution, which implies that the shortcomings originate from the inherent contrastive learning mechanism. Besides, existing advanced GCL models inherit these frameworks, so we think these issues still exist in most GCL models, which could be demonstrated in the quantitative analysis that the advanced models present unideal GMs and TINs.
>
> **Q3.**
>
> According to the statement in Section 4.3, it is **a general condition** that a larger “Eps” leads to fewer Mean Groups (MGs), but there also exist special cases where a larger “Eps” leads to great MGs since the “Eps” is too small to form groups. In other words, the DBSCAN algorithm is hard to detect valid groups since the minimal distance between the sparsely scattered samples is greater than “Eps”, and these samples are finally viewed as isolated samples. This can be demonstrated by the results that the “MGs” of GREET with “Eps=0.5” is higher than that with “Eps=0.7”, but the number of total isolated samples (TINs) of GREET with “Eps=0.5” is also more than that with “Eps=0.7”. Therefore, it also inspires us to evaluate the models by comprehensively considering both MGs and TINs but not only one of them. That’s also our motivation to propose the two metrics for evaluating global sparseness and local cohesion.
>
> **Q4.**
>
> We select SOTA methods for a fair comparison. However, they fail to generate an ideal feature distribution for local cohesion and global sparseness. Including outdated methods may reduce these insights due to performance gaps caused by weaker architectures rather than inherent shortcomings. We highlight that even the SOTA models face the challenge to reinforce the necessity of our model, and a similar comparison method is adopted by [1].
>
> [1]Augmentation-free self-supervised learning on graphs, in AAAI 2022.
>
> **Q5.**
>
> Negative correlation learning in ensemble methods encourages diversity among individual learners to enhance overall performance. Similarly, the fuzzy boundary construction introduces several different Gaussian membership functions to make each cluster (individual learner) display flexibility and diversity for fitting various samples. This process prevents over-concentration within local groups and enhances the model ability to capture a more globally structured distribution.
>
> A direct analogy can be drawn to ensemble methods that introduce perturbations or penalize excessive similarity between learners, which in our model is to apply the prototypical contrastive loss for enlarging the distances between different clusters. The exploration for regularization techniques from ensemble learning could inspire additional improvements in optimizing fuzzy boundary formation.

---

### Official Review · Reviewer_Ytiy · 2025-02-25

**Overall Recommendation:** 4

**Summary:**

To address the ingrained shortcomings such as local cohesion and global sparseness in graph contrastive learning (GCL), this paper introduces a novel GCL model incorporating fuzzy boundaries. The proposed method dynamically extends original cluster boundaries to mitigate the two shortcomings in the embedding space. Extensive experiments on downstream tasks and quantitative analysis demonstrate that the proposed model achieves superior performance when compared with state-of-the-art methods.

**Claims And Evidence:**

The evidence such as the t-sne visualization of current graph contrastive learning models could support the problem claims. The method design and the experiment results support the claims that the proposed model could mitigate the shortcomings.

**Essential References Not Discussed:**

Current references can help understand the paper and the comparative models are also new, but comparisons with self-supervised graph generative learning models are also essential.

**Experimental Designs Or Analyses:**

The experimental designs to realize comparative experiments are sound. However, the analyses or the explanations for the experiments are not well detailed, such as the definitions of group, group list, and isolated nodes, which are important for the readers to understand.

**Methods And Evaluation Criteria:**

This proposed method combines the graph contrastive model with the fuzzy boundary in order to extend the original cluster boundaries to fill the discrete and sparse cluster distributions. The method seems to make sense.

**Other Comments Or Suggestions:**

See the above.

**Other Strengths And Weaknesses:**

**Strengths**

1. Clear structure. The paper is well-written and logically organized, with a clear problem statement, detailed methodology, and thorough experimental analysis.

2. Sound technique. The proposed fuzzy boundary that expands the original cluster boundary with fuzzy set theory appears technically sound. The experiments in different tasks show the comparative performance with graph contrastive models.

**Weaknesses**

1. Limited model diversity in baselines. The selected baseline models mainly focus on contrastive learning methods. To provide a more comprehensive evaluation, the paper could benefit from including comparisons with graph generative models.

2. Troublesome hyperparameter tuning. The proposed method adopts different hyperparameters for various datasets. And sensitive hyperparameters such as the penalty coefficient and temperature coefficient may affect the model performance, which are troublesome to choose for different datasets.

**Questions For Authors:**

1. Quantitative analysis. Hard to understand the concepts such as "Group" and "Group List" in the quantitative analysis, and it needs to provide clearer and more concrete explanations of them.

2. Ablation study. A reasonable ablation study should test the model without pre-training, which can better reveal the function of the pre-training process.

3. Fuzzy boundary design. The proposed fuzzy boundary seems to be only used for expanding the original cluster boundary but works well. Do the fixed hyperparameters work well or better than the fuzzy boundary?

----------------------

Thank you  for your careful responses. All my concerns have been well addressed and I strongly suggest you to add these new results into the final version. I am satified with your answer and decide to raise my scores.

**Relation To Broader Scientific Literature:**

This paper first finds two shortcomings in graph contrastive learning, but I am not sure whether the problems are significant for research on graph contrastive learning.

**Theoretical Claims:**

I did check all the theoretical claims and most proofs except for the proof on the model generalization. These theoretical claims and proofs are easy to understand.

---

> ### Author Rebuttal · Authors · 2025-03-30
>
> **W1.**
>
> We compare our model with two advanced graph generative models, S2GAE [1] and Bandana [2]. S2GAE reconstructs randomly masked edges while Bandana samples edge masks from a continuous distribution. As shown in Table 1, these graph generative models perform competitively but are limited to small datasets. On large datasets like Ogbn_arxiv, their performance declines due to graph sparsity that limits topology-based knowledge extraction. In contrast, our proposed model can connect and tighten similar samples in the representation space by introducing the fuzzy boundary, which effectively explores the commonalities regardless of dataset size.
>
> Table 1
>
> |Models|Cora|Citeseer|Pubmed|Photo|Computers|DBLP|Arxiv|
> |-|-|-|-|-|-|-|-|
> |S2GAE|85.5±0.3|73.4±0.5|86.5±0.1|93.5±0.2|90.0±0.1|84.1±0.0|60.9±0.0|
> |Bandana|85.5±0.8|73.0±0.8|86.4±0.3|93.4±0.1|89.6±0.1|83.9±0.0|66.8±0.0|
> |Ours|**85.9±0.6**|**74.1±0.6**|**87.0±0.3**|**93.7±0.2**|**90.8±0.2**|**84.8±0.2**|**69.5±0.1**|
>
> [1]S2GAE: Self-supervised graph autoencoders are generalizable learners with graph masking, in WSDM 2023.
>
> [2]Masked graph autoencoder with non-discrete bandwidths, in WWW 2024.
>
> **W2.**
>
> Section B.4 shows that the hyperparameters such as temperature coefficient and penalty coefficient are not sensitive and don’t impact the performance significantly for a given dataset.
>
> Additionally, it is standard to adopt dataset-specific hyperparameters in graph representation learning due to variations in graph properties and sizes, which is easily found in numerous literature [1,2]. For a small dataset like Amazon Photo (a co-purchase network) and a large dataset like Ogbn-arxiv (a citation network), their differences should be addressed with subtle hyperparameter settings. However, for similar datasets, hyperparameters transfer easily, which facilitates the hyperparameter search process. For example, the Amazon Photo and Amazon Computers datasets have similar graph properties (co-purchase networks) and dataset size (this condition also applies to Cora and Citeseer datasets), so the hyperparameters for them are usually similar. It also means that it won’t spend much overhead when knowing more about the dataset properties. Thus, hyperparameter tuning is reasonable and manageable.
>
> [1]Homogcl: Rethinking Homophily in Graph Contrastive Learning, in KDD 2023.
>
> [2]A New Mechanism for Eliminating Implicit Confict in Graph Contrastive Learning, in AAAI 2024.
>
> **Q1.**
>
> We provide concrete definitions of the terms: (1) A **Group** is a set of nodes that are close to each other in the embedding space, with the maximal distance between each node pair smaller than pre-defined “Eps”, where “Eps” is a hyperparameter of DBSCAN to limit the maximal distance between the representations in a detected local group. (2) A **Group list** is a list containing the number of groups detected within each class cluster, with each element in the list corresponding to the number of groups in a given cluster. (3) **Isolated nodes** are the nodes whose distances between themselves and any other group are greater than “Eps”, and these nodes are not assigned to any group and contribute to global sparseness.
>
> **Q2.**
>
> To ensure rigor, we supplement Table 2 with results for models without pre-training. Comparisons show that pre-training significantly improves performance, especially on small datasets like Cora. This confirms that pre-training helps extract critical features from the data and gather samples with great similarities to form reliable clusters.
>
> Table 2
>
> |Pre-training|Cora|DBLP|Photo|Computers|
> |-|-|-|-|-|
> ||84.86±0.58|84.35±0.14|93.53±0.19|90.46±0.22|
> |√|**85.90±0.56**|**84.80±0.17**|**93.71±0.20**|**90.75±0.20**|
>
> **Q3.**
>
> Fuzzy boundaries **dynamically adapt to evolving cluster structures**, unlike fixed hyperparameters which fail to adjust to data distribution changes. Since clusters become tight enough during training, the pre-defined hyperparameters may be too large and prone to including samples from other clusters, which degrades the performance quickly.
>
> Besides, Fixed hyperparameters also struggle with feature dimensions of varying scales. Our fuzzy boundary with learnable membership functions flexibly adapts to different feature scales.
>
> Finally, we also provide the experiment results to compare the fuzzy boundary (represented as “Fuzzy”) with the fixed margin expansion in Table 3, which confirms that models with fixed margins require fine-tuning yet still underperform compared with our model, demonstrating the difficulty of using static hyperparameters in a dynamic training process.
>
> Table 3
>
> |$\epsilon$|Cora|Citeseer|DBLP|Arxiv|Photo|Computers|
> |-|-|-|-|-|-|-|
> |0.05|85.6|73.5|84.3|69.4|93.6|90.5|
> |0.1|85.6|73.0|84.5|69.4|93.5|90.5|
> |0.15|85.5|72.8|84.3|69.4|93.5|90.3|
> |0.2|85.4|72.5|84.3|69.3|93.6|90.1|
> |0.25|85.3|72.6|84.2|69.3|93.2|90.2|
> |0.3|85.1|72.3|84.0|69.2|93.1|90.3|
> |Fuzzy|**85.9**|**74.1**|**84.8**|**69.5**|**93.7**|**90.8**|

---

### Official Review · Reviewer_7eQp · 2025-03-02

**Overall Recommendation:** 3

**Summary:**

This paper presents a novel graph model by introducing fuzzy set theory to alleviate the impacts of local cohesion and global sparseness, expecting to form the ideal global structure distribution. The proposed model realizes a learned boundary to bridge the local groups and isolated samples by adjusting the cluster boundary adaptively. The experiments across several datasets validate the model performance. The quantitative analysis and visualization demonstrate the model generates better global structural distribution.

**Claims And Evidence:**

The Introduction clarifies the problems with visualization in Figure 1 and gives the explanations. Experiment results and quantitative analysis illustrate that the model alleviates the problems. These evidences can support the claims in the submission.

**Essential References Not Discussed:**

The paper adopts a pre-training and fine-tuning learning framework in graph contrastive learning, which is similar to a previous paper on graph contrastive clustering [1], so it is better to cite it.


[1] Dink-net: Neural clustering on large graphs. In International Conference on Machine Learning 2023.

**Experimental Designs Or Analyses:**

These experiments evaluate the proposed model based on downstream tasks and quantitative analysis. The experiment setting in node classification and node clustering are common, and the evaluation metrics are widely adopted. Additional quantitative analysis also shows the model ability to solve the shortcomings. The experimental designs are sound and convincing.

**Methods And Evaluation Criteria:**

The proposed method designs a learnable boundary to involve the isolated samples belonging to the cluster with great similarities and local groups near the cluster boundary. By contracting the fuzzy boundary, the samples within the fuzzy boundary are gathered more tightly to alleviate local cohesion and global sparseness. These processes make sense for solving the problem.

**Other Comments Or Suggestions:**

I hope the authors can tackle my weaknesses and questions.

**Other Strengths And Weaknesses:**

Strengths
(1) This paper focuses on mitigating local cohesion and global sparseness in graph contrastive learning, which corrupts global structure distributions, and the problem is meaningful.
(2) This paper proposes fuzzy boundary construction to mitigate local cohesion, and fuzzy boundary contraction to alleviate global sparseness. The method is effective in restoring the ideal global structure distribution.
(3) Besides promising model performance in two tasks, this paper also adopts quantitative analysis to quantify the model ability to mitigate the shortcomings when compared with other graph models.

Weaknesses
(1) When centralizing the node representations, the model utilizes node representation $H$ to subtract the corresponding prototypes $C$. However, the matrix size of $H$ is $N \times d$ while the matrix size of $C$ is $C_p \times d$. The authors should detail more about the centralization process.
(2) The view angle for Figure 4(a) may be not fair. Though Figure 4(b) has better visualization than Figure 4(a), it may be attributed to the fact that Figure 4(b) is observed from a better view angle. The authors could give a more fair angle for Figure 4(a).

**Questions For Authors:**

(1) Equation 9 needs more detailed explanations. For example, how does it work to realize intra-cluster compactness, and why does higher temperature coefficient $\tau$ weigh the intra-cluster compactness more?
(2) The motivation of this paper is to generate a global structure distribution that realizes intra-cluster compactness and inter-cluster separateness. However, the fuzzy boundary contraction only focuses on realizing intra-cluster compactness intuitively. How does the model realize high inter-cluster separateness?

**Relation To Broader Scientific Literature:**

This paper first attends to two ingrained shortcomings in GCL, which could cause inferior global structural distribution. Then the paper proposes a novel idea to expand cluster boundaries to bridge the discrete local groups and isolated samples. Experiment results
 demonstrate the model could alleviate the shortcomings. This paper may contribute to the broader scientific literature to some degree.

**Theoretical Claims:**

I checked all the theoretical claims and proofs, including boundary construction and model generalization, which made the model design more reasonable.

---

> ### Author Rebuttal · Authors · 2025-03-30
>
> **W1.**
>
> The details of the centralization are as follows:
>
> When centralizing the node representations, we use the K-means clustering algorithm to obtain the cluster prototype $C\in\mathbb{R}^{C_p\times d}$, where $C_p$ is the number of clusters, and $d$ is the dimension of the prototype vector. For each node $v_i$ belonging to a specific cluster $k$, we assign it to its corresponding prototype $c_k$ and perform the centralization as follows:
>
> $z_i = h_i - c_k$
>
> where $h_i$ is the original node representation and $c_k$ is the prototype of the cluster that node $v_i$ belongs to. We also provide a matrix form by constructing an assignment matrix $M \in \mathbb{R}^{N\times C_p}$, where $M_{ik}=1$ if node $v_i$ belongs to cluster $k$, and 0 otherwise. Then, the centralized representation can be expressed as:
>
> $Z=H-MC$
>
> where $MC$ selects the corresponding prototype for each node. This modification with the matrix form ensures the dimensions are consistent during the centralization process.
>
> **W2.**
>
> The visualization of Figure 4(a) is affected by the choice of view angle. To ensure a fair comparison, we have updated the visualization with a new version of Figure 4(a) using a more ideal view angle and supplemented the further analysis in the refined paper. Due to inconvenience in providing images in the rebuttal phase, we only provide the analysis as follows:
>
> For the adjusted angle in Figure 4(a), only three clusters become clearer and the embedding distribution in Figure 4(a) is inferior to Figure 4(b) since most samples in different classes are overlapped together, which further verifies that the fuzzy boundaries have the ability to refine the embedding distribution.
>
> **Q1.**
>
> Equation 9 aims to tighten samples in a cluster while pushing samples away from different clusters. Specifically, the loss encourages each node representation $h_i$ to be close to its corresponding cluster prototype $p_i$. A lower distance $\||h_i-p_i\||^2_2$ results in a higher probability that pushes representations toward their prototypes, thus tightening clusters.
>
> For better understanding Equation 9, we use a softmax-based probability function that $v_i$ belongs to the corresponding prototype $p_i$ as:
>
> $P(i)=\frac{e^{-\tau\||h_i-p_i\||^2_2}}{\sum_{j=1}^{C_p}e^{-\tau\||h_i-p_j\||^2_2}}=\frac{1}{\sum_{j=1}^{C_p}e^{\tau(\||h_i-p_i\||^2_2-\||h_i-p_j\||^2_2)}}$
>
> where $h_i$ is the representation of node $v_i$, $\tau$ is the temperature coefficient that scales the Euclidean distances. According to the assumption that $v_i$ belongs to the corresponding prototype $p_i$, $\||h_i-p_i\||^2_2-\||h_i-p_j\||^2_2$ is negative in general since $p_j$ is not the corresponding prototype of $h_i$, so $\||h_i-p_j\||^2_2\geq\||h_i-p_i\||^2_2$. A larger $\tau$ will make the denominator smaller, and $P(i)$ becomes larger accordingly, which implies the cluster becomes more compact.
>
> **Q2.**
>
> Although fuzzy boundary contraction mainly focuses on intra-cluster compactness, the model also ensures inter-cluster separateness through the prototype contrastive loss (Equation 9). The key factor is that the prototype contrastive loss adjusts the distances between the node representation and the prototypes. The training process with Equation 9 pulls the representations and their corresponding prototypes together and pushes representations away from non-corresponding prototypes $p_j \(j\neq i\)$, which **explicitly increases inter-cluster distances**.
>
> Additionally, the fuzzy boundaries also **implicitly enhance inter-cluster separateness**, which is explained in Figure 6(b) and Figure 6(c). Concretely, the fuzzy clusters include both real samples (also forming the inner real clusters) and fuzzy boundary elements that are always outside the real samples. When the training process ends, the virtual fuzzy boundary elements are removed and the distances between the inner real clusters are further enlarged. As a result, the clusters embody higher intra-cluster compactness and inter-class separateness merits.
>
> Thus, the combination of fuzzy boundary and prototype contrastive loss results in both intra-cluster compactness and inter-cluster separateness, achieving the desired global structural distribution.

---

> > ### Comment · Reviewer_7eQp · 2025-04-04
> >
> > I appreciate the authors' responses, which have effectively resolved my initial concerns. Having carefully reviewed the other comments, I maintain my positive assessment.

---

> > > ### Author Response · Authors · 2025-04-04
> > >
> > > We sincerely appreciate your constructive feedback and valuable suggestions, which have greatly helped us improve our work. Thank you for recognizing our efforts in addressing your concerns—we are truly grateful for your time and insightful comments throughout the review process.

---

### Official Review · Reviewer_upat · 2025-03-10

**Overall Recommendation:** 3

**Summary:**

This paper addresses two integrated shortcomings in graph contrastive learning, namely local cohesion and global sparseness, which will lead to inferior global structural distributions. So this paper proposes novel fuzzy boundaries to gather discrete local groups and isolated samples, which effectively alleviate the local cohesion and global sparseness. Overall, this paper treats a detailed problem in graph contrastive learning, and the method is interesting.

**Claims And Evidence:**

Yes, the claims are supported by clear and convincing evidence. The evidence such as the performance on downstream tasks and visualization illustrate that the problems are well addressed.

**Essential References Not Discussed:**

This paper adopts prototype learning in the method. However, the paper lacks the related works of graph prototype learning such as [1].

[1] X-GOAL: Multiplex heterogeneous graph prototypical contrastive learning. In CIKM.

**Experimental Designs Or Analyses:**

Yes, the experimental designs are sound to validate the model ability. Specifically, the paper adopts node classification, node clustering, ablation study, quantitative experiments, and a series of visualizations to analyze the model.

**Methods And Evaluation Criteria:**

Yes, the proposed fuzzy boundary makes sense for the problem and addresses the problem well. It firstly expands the cluster boundary with the fuzzy set theory to gather discrete local groups and isolated samples, then contracts the boundary to tighten samples within the boundary, which effectively alleviates local cohesion and global sparseness.

**Other Comments Or Suggestions:**

See weakness.

**Other Strengths And Weaknesses:**

**Strengths**

++ Good writing and clear organization. The paper has good structural organization and a clear problem statement, which is easy to read and understand.

++ Interesting method. This proposed model introduces the novel idea with fuzzy set theory to expand the original cluster boundary, which mitigates the local cohesion and global sparseness.

++ Sufficient experiments. The experimental design is comprehensive and convincing by covering different downstream tasks, quantitative analysis, and visualization, which well demonstrates the model effectiveness.

**Weaknesses**

-- This paper adopts prototype learning in the method. However, the paper lacks related works on graph prototype learning, it would be better to provide a review of graph prototype learning in related works.

-- The icons of local groups in Figure 2 is misleading. In my opinion, the samples in local groups are not isolated samples, but they are depicted as isolated samples in Figure 2, and the authors should correct the figure.

**Questions For Authors:**

1. I have a minor question regarding the pre-training visualization. The authors mentioned that projecting raw data into a 3-dimension space can lead to information loss. However, I think projecting it into a 2-dimension space might result in more information loss and cause worse visualizations. Despite this, Figure 4 shows poor results in 3-dimension space, while Figure 6 demonstrates a more regular distribution in the 2-dimension visualization after pre-training. Could you provide a brief explanation for this phenomenon?

**Relation To Broader Scientific Literature:**

This paper focuses on addressing the shortcomings commonly emerging in the current research, and the results demonstrate that the proposed model could effectively address the shortcomings.

**Theoretical Claims:**

Yes, the theoretical claims are relative to the proposed model. I checked the proofs for theoretical claims, and did not find obvious issues.

---

> ### Author Rebuttal · Authors · 2025-03-30
>
> **W1.**
>
> Following your kind suggestion, we provide a related work compassing graph prototype learning as follows:
>
> Graph prototypical learning has emerged as a significant direction in graph representation learning, addressing limitations in traditional contrastive and few-shot learning approaches. Existing GCL methods are limited to sub-optimal representations due to sampling bias where semantically similar graphs are incorrectly treated as negatives. To overcome this, PGCL introduces a clustering-based approach by grouping semantically similar graphs and ensures that negative samples come from different clusters [1]. Similarly, GPCL [2] enhances GCL by integrating both node-level and prototype-level contrastive objectives, maintaining consistency across different augmentations. To further capture global semantic structures, GraphLoG [3] employs K-means clustering to derive hierarchical prototypes. At the same time, X-GOAL [4] extends prototypical learning to multiplex heterogeneous graphs for aligning embeddings across different relational layers. In the domain of few-shot learning, GPN [5] leverages meta-learning to extract transferable prototypes for node classification with scarce labels. Similarly, S4GPN [6] integrates self-supervised learning with prototypical networks to improve hyper-spectral image classification. These advancements demonstrate the effectiveness of prototype-based graph learning in mining cluster-level semantic consistency which is important for realizing ideal global structural distribution.
>
> [1]Prototypical graph contrastive learning, in TNNLS 2024.
>
> [2]Graph prototypical contrastive learning, in INS 2022.
>
> [3]Self-supervised graph-level representation learning with local and global structure, in ICML 2021.
>
> [4]X-GOAL: Multiplex heterogeneous graph prototypical contrastive learning, in CIKM 2022.
>
> [5]Graph prototypical networks for few-shot learning on attributed networks, in CIKM 2022.
>
> [6]Self-supervised spectral-spatial graph prototypical network for few-shot hyperspectral image classification, in TGRS 2023.
>
> **W2.**
>
> Thanks for your careful observation regarding Figure 2. To address this issue, we have revised Figure 2 by (1) adopting a different line shape to represent the local groups, and (2) replacing the samples inside the local groups with crisp representations. The updated model architecture will be provided in the refined paper to avoid misinterpretation.
>
> **Q1.**
>
> The difference in visualization between the two figures is attributed to different data processing. For realizing the 3-dimension visualization in Figure 4, we first condense the raw data into a 3-dimension feature space with graph auto-encoder, then use the crisp and the proposed fuzzy models to train the 3-dimension features for visualization. Since the raw data is condensed into a low-dimension feature space from a high-dimension space, the data information is heavily lost, which presents terrible visualization. Even so, the proposed fuzzy model could also gather similar samples in the clusters, which leads to much clearer decision cluster boundaries when compared with the crisp model.
>
> The visualization in Figure 6 is general t-sne. We first apply the proposed model to train the raw graph data, and then implement the t-sne algorithm on the produced representations. In this case, the models could preserve more critical information in the feature space than in the 3-dimension space, which makes Figure 6 have much better visualization than Figure 4.

---

### Decision · Program_Chairs · 2025-05-01

**Decision:**

Accept (poster)

**Comment:**

This paper introduces a novel fuzzy boundary mechanism for graph contrastive learning (GCL), addressing two key shortcomings: local cohesion (discrete local groups within clusters) and global sparseness (isolated samples). The proposed method dynamically adjusts cluster boundaries using fuzzy set theory, improving intra-cluster compactness and inter-cluster separability. Experiments on node classification and clustering tasks demonstrate superior performance over state-of-the-art GCL methods.

The paper is with good theoretical grounding (fuzzy boundary construction/proofs) and comprehensive experiments (ablation, sensitivity analysis). The paper makes a contribution to GCL by addressing underexplored shortcomings with a theoretically grounded, empirically validated approach. While computational overhead and reliance on pre-training are noted limitations, all reviewers think that the rebuttal strengthened the paper’s rigor and scope.